# Computational signatures of uncertainty are reflected in motor cortex excitatory neurochemistry

Nazia Jassim [1,2] ✉, Peter Thestrup Waade[3], Owen Parsons[2], Frederike H. Petzschner[4], Catarina Rua [5,6], Christopher T. Rodgers [5], Simon Baron-Cohen [2], John Suckling [2], Christoph Mathys [3] & Rebecca P. Lawson [1]

How individuals process and respond to uncertainty has important implications for cognition and mental health. Here, we use computational phenotyping to examine inter-individual differences in uncertainty processing in relation to neurometabolites and trait anxiety in humans. We introduce a categorical state-transition extension of the Hierarchical Gaussian Filter to model individuals' evolving beliefs about transition probabilities in a four-choice probabilistic sensorimotor learning task with a reversal. Using 7-Tesla Magnetic Resonance Spectroscopy, we measure neurotransmitter levels in the primary motor cortex. Model-based results reveal dynamic belief updating in response to environmental changes. We further find region-specific relationships between baseline primary motor cortex glutamate+ glutamine levels and prediction errors and volatility beliefs. High trait anxiety is associated with faster post-reversal responses. This study establishes a direct neurochemical correlate of hierarchical belief updating, identifying motor cortex glutamate + glutamine as an important neural marker of inter-individual differences in uncertainty processing.

Navigating uncertain environments presents a unique challenge to the human brain, and by extension, mental health. The ability to extract statistical regularities from the sensory environment is of paramount importance to cognition and overall development[1–4]. This is considerably more challenging in unpredictable or volatile environments. Here we are required to constantly adapt to the changing statistics and to update our expectations based on unexpected outcomes[5–7]. To deal with this challenge, we estimate different types of uncertainty and use these estimates to guide our actions[8–10]. When faced with similar contexts, individuals may process and respond to uncertainty differently, suggesting differences in their beliefs or estimates of how uncertain the environment is[7,11,12]. Altered learning under uncertainty has been found to predict mental health difficulties, especially

anxiety[13–22]. Consequently, it is crucial to not only understand the neurocomputations of uncertainty processing but also to use computational phenotyping[23–26] to identify hidden variables that account for individual differences.

Emerging evidence suggests that different types of uncertainty, such as expected uncertainty (known variability of outcomes), unexpected uncertainty (sudden deviations from expected patterns), and volatility (the changeability of the environment over time), may be represented by partially distinct neural mechanisms and computational processes[7,8,10–12]. Volatility, a higher-order form of uncertainty, reflects individuals' inferred uncertainty about potential changes in the environment. In an uncertain environment, individuals track volatility and adjust their learning accordingly, weighting new information more

[1]Department of Psychology, University of Cambridge, Cambridge, UK. [2]Autism Research Centre, University of Cambridge, Cambridge, UK. [3]Interacting Minds Centre, Aarhus University, Aarhus, Denmark. [4]Carney Institute for Brain Science, Brown University, Providence, RI, USA. [5]Wolfson Brain Imaging Centre, University of Cambridge, Cambridge, UK. [6]Perceptive Discovery, London, UK. ✉e-mail: nj304@cam.ac.uk

heavily when volatility is perceived to be high, and more conservatively when volatility is low[5,10–12,27]. Individual differences in these computations have been linked to trait anxiety, with earlier studies suggesting that more anxious individuals may overestimate volatility or update beliefs less adaptively[14,28]. However, recent large-sample studies have not consistently replicated these findings[21,29,30], highlighting the need for further investigation. Uncertainty processing has also been linked to neuromodulatory systems, including acetylcholine, noradrenaline, dopamine, and glutamate[7,12,31–33]. Therefore, it is important to employ mechanistic models that can dissociate and quantify these distinct forms of uncertainty in order to better understand their neural substrates and relevance for mental health.

While there has been significant progress in the computational modelling of adaptive learning and choice behaviours, experimental paradigms and models have mainly been limited to binary choices and outcomes. However, in the real world, learning and decision-making under uncertainty are far more complex. In their seminal work on the multi-armed bandit task, Daw et al. (2006) showed that presenting individuals with four options – as opposed to two – significantly increased the levels of uncertainty, prompting more complex strategies for tracking and adapting to changing reward contingencies[34]. In a similar vein, but moving beyond reward-based learning, we examine how individuals implicitly learn the probabilistic structure of a four-choice sensorimotor learning task and how their beliefs about volatility shape dynamic learning. While studies have found that volatility increases learning[5,10–12,27], the complex interplay between experimentally induced uncertainty and non-binary choices is unclear. To account for this, we introduce an instantiation of a hierarchical Bayesian model for adaptive learning that captures how people implicitly build and update beliefs about transition probabilities across four possible options.

The Hierarchical Gaussian Filter (HGF) is a Bayesian model of adaptive learning that estimates hidden, dynamically changing states of the environment based on noisy observations[6,35]. These estimates correspond to probabilistic "beliefs" across multiple hierarchical levels: the lowest level represents raw sensory input, while higher levels capture more abstract beliefs about the structure and volatility of the environment. In its original formulation, the HGF employed volatility coupling, wherein higher levels modulate the rate of change (i.e., learning rate) at lower levels[6]. However, classic predictive coding assumes that higher levels predict the value of lower levels (i.e, value coupling). To formalise this distinction, the recent generalised version of the HGF introduces a network-based formulation that allows for both volatility coupling and value coupling between hierarchical belief states[36]. In this extended framework, each belief is represented as a probabilistic node, and higher-level nodes can influence both the drift (value) and volatility (rate of change) of lower-level nodes, enabling a richer and more flexible modelling of adaptive learning[36].

The brain uses probabilistic models to optimise performance and guide responses during learning[37–39]. Laboratory-based tasks of implicit probabilistic learning are designed to probe how participants acquire the underlying probabilistic structure without explicit awareness. In the present study, participants perform a sensorimotor task involving two sequences that differ in their probability of occurrence, with one occurring frequently and the other less so (Fig. 1). The task examines how individuals implicitly learn the underlying probabilistic structure, using motor output as a proxy for learning performance. As sensory evidence is accumulated, it is prominently represented in motor cortical areas to guide task-relevant actions[40,41]. At the cellular level, as learning progresses, primary motor cortex neurons (M1) neurons undergo adaption and glutamate-driven plasticity[42,43]. Specifically, the primary excitatory neurotransmitter glutamate plays a critical role in rapid inter-cellular communication to facilitate learning and choice behaviour[44]. Research in rodents has shown that learning in dynamic environments—as measured through reversal learning

paradigms—is strongly dependent on glutamatergic modulation[45–48]. With the advent of Magnetic Resonance Spectroscopy (MRS), a non-invasive technique to measure tissue metabolites in vivo, the metabolites related to probabilistic learning can now be studied effectively in living humans[49,50].

Here, we use a combination of experimental psychology, computational cognitive modelling, and brain imaging to examine uncertainty processing in relation to neurochemistry in humans. We capitalise on the heightened sensitivity of 7-Tesla (7 T) MRS to measure excitatory glutamate + glutamine (Glx) and inhibitory γ-aminobutyric acid (GABA) in the primary motor cortex (M1). To examine uncertainty processing, we implement a four-choice probabilistic sensorimotor learning task with a reversal to gauge how individuals implicitly learn the underlying statistics of the task, and how this learning is updated following a switch to the probabilities (Fig. 1). As the learning is implicit, it can be gauged through motor responses. We further probe the latent variables and hidden states underpinning task performance by means of computational modelling. Notably, we introduce the categorical state-transition HGF to model probabilistic learning as—assessed by reaction times—in the well-established four-choice Serial Reaction Time (SRT) task[3] (Fig. 1). We further examine the relationships between computational model-based parameter estimates of sensorimotor probabilistic reversal learning and neurotransmitter levels. Finally, we test the hypothesis that beliefs about volatility are dependent upon trait anxiety levels.

Our findings demonstrate that participants implicitly learned the probabilistic structure of the task, evidenced by the speeding of responses on high versus low probability trials as the task progressed (Fig. 2). Post-error slowing indicated that errors played a key role in driving learning (Fig. 2). These model-agnostic results were further supported by computational modelling (Fig. 3) findings, which suggest that learning was driven by surprise and beliefs about volatility (Fig. 4). Importantly, we found strong correlations between participants' M1 glutamate + glutamine (Glx) levels and prediction errors and volatility beliefs (Fig. 5). This suggests that M1 Glx plays a crucial role in high-level volatility beliefs during implicit learning under uncertainty. We also found that, while individuals high in trait anxiety were faster after the reversal, contrary to our predictions, there was no association between trait anxiety and volatility beliefs. Our findings offer insights into the computations underlying probabilistic learning and reveal important neurochemical correlates of adaptive behaviour in unpredictable environments.

## Results
### Implicitly-learned probabilities and errors drive performance
We implemented a probabilistic serial reaction time (SRT) task in which participants ($n = 42$) were told to indicate the location of a stimulus by pressing keys corresponding to four possible locations (Fig. 1a). Unbeknownst to the participants, the stimulus location on each trial was associated with pre-determined probabilities that may be implicitly learnt over time (Fig. 1b–d, Methods). Following one session of the task, there was a switch to the probabilities to gauge the impact of a probabilistic reversal on implicit learning.

To further examine whether the probabilistic sequences were implicitly learned, we fit linear-mixed effects (LME) models to the data. We examined the main effects of stimulus probability (High vs Low Probability) and session (Pre- vs Post-reversal) on log-transformed RT (LME model 1). This indicated main effects of both stimulus probability ($t(79,200) = 2.48$, $p = 0.013$, $b = 0.004$, $SE = 0.002$, 95% CI [0.001,0.008]) and session ($t(79,200) = -7.41$, $p = 1.28 \times 10^{-13}$, $b = -0.009$, $SE = 0.002$, 95% CI [−0.011, −0.007]) (Fig. 2a).

To examine learning on a finer timescale, we also modelled trial-by-trial changes in log-transformed RT (LME Model 2). This model included fixed effects for stimulus probability (High vs Low Probability), session (Pre- vs Post-reversal), trial number, as well as all

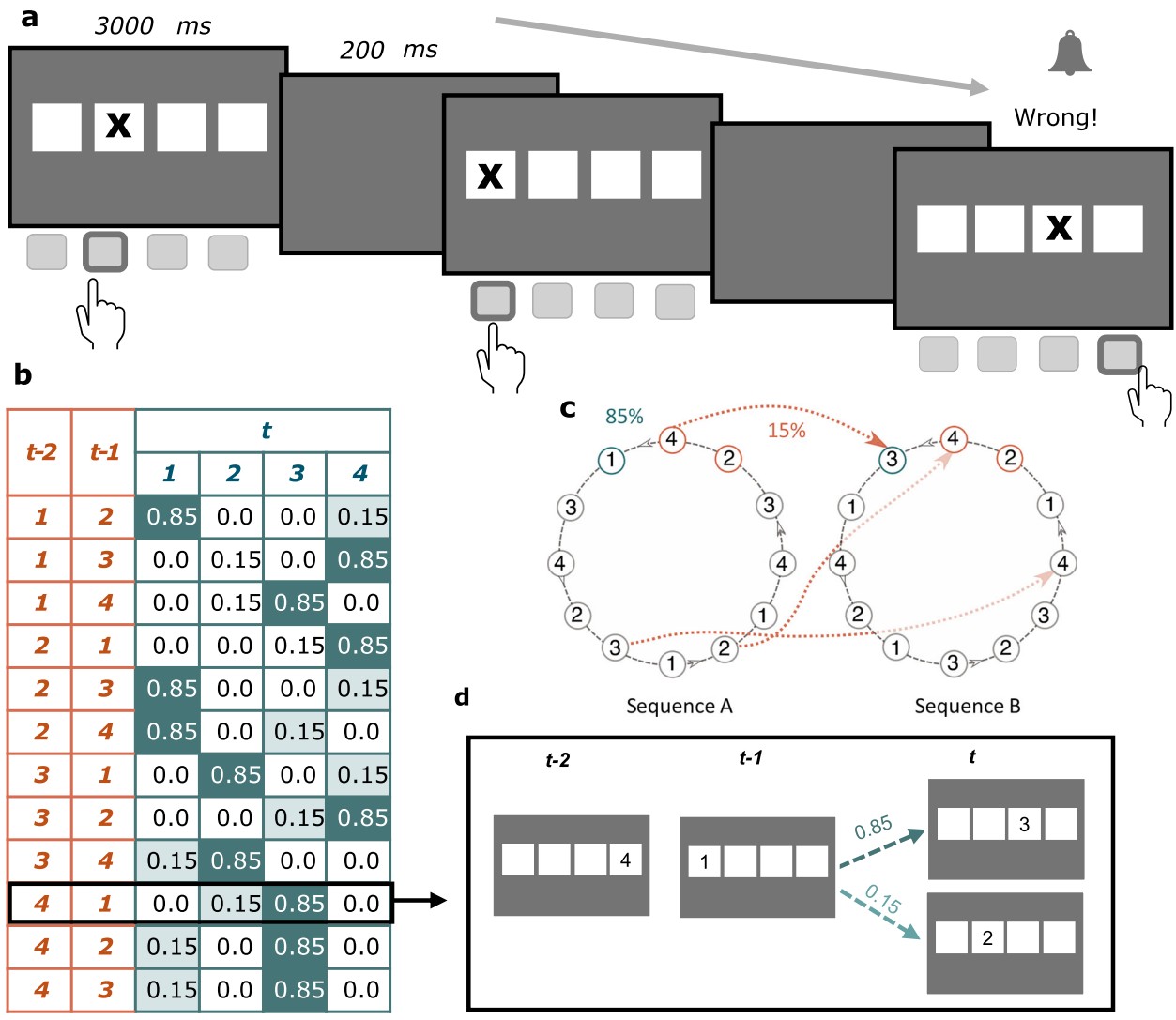

**Fig. 1 | Probabilistic Serial Reaction Time task. a** Schematic representation of the stimuli and paradigm. Participants were instructed to indicate the location of the stimulus by means of a button press corresponding to each of the 4 possible locations. The task consisted of two sessions (Pre- and Post-reversal), comprising 960 trials each. **b** Transition probabilities of the stimulus locations. The location of the stimulus on each trial was decided by two Markov chain sequences. The two sequences differed in the second-order conditional occurrence, with one sequence occurring 85% of the time and the other 15%, leading to High and Low Probability trials (indicated by dark and light green shading, respectively). **c** The two deciding sequences, with example transitions between sequences. Here, Sequence A (85%) has a high probability of occurring, while Sequence B has a low probability of occurring (15%). Following one session (960 trials) of the task, the probabilities of each sequence occurring were reversed, leading to Sequence A as the low-probability sequence and Sequence B as the high-probability one. **d** An example transition leading to High and Low Probability trials.

interactions. Crucially, it also included random intercepts and random slopes for trial number by participant to account for inter-individual differences in learning trajectories. We found a significant interaction between stimulus probability and trial number. Specifically, for low probability stimuli, responses slowed over time compared to high probability stimuli ($t(79,160) = 5.22$, $p = 1.83 \times 10^{-7}$, $b = 0.009$, $SE = 0.002$, 95% CI [0.006, 0.013]). The interaction between session and trial number was also significant ($t(79,160) = 13.06$, $p = 2 \times 10^{-16}$, $b = 0.018$, $SE = 0.002$, 95% CI [0.015,0.019]). These findings suggest that participants differed meaningfully in their trial-by-trial learning and adjusted their responses based on both stimulus probability and task session. A more detailed visualisation of the temporal effects can be found in Fig. 2b.

Overall, accuracy rates were high across the pre-reversal (*Mean* = 0.92, *SD* = 0.20) and post-reversal (Mean=0.93, SD = 0.20) sessions. We examined whether trial-by-trial errors had an effect on probabilistic learning by implementing an LME model on log-transformed RT with outcome of the previous trial (Correct vs Wrong) and session (Pre- vs Post-reversal) as predictors (LME Model 3). We found a main effect of errors on response speed of subsequent trials ($t(79,200) = 19.67$, $p = 2 \times 10^{-16}$, $b = 0.11$, $SE = 0.006$, 95% CI [0.099,0.120]). In addition, we found an interaction between post-error trials and session ($t(79,200) = -2.02$, $p = 0.04$, $b = -0.016$, $SE = 0.008$, 95% CI [−0.031, −0.0004]), suggesting a reduced effect of post-error slowing in the post-reversal session (Fig. 2 c-d).

**Categorical state-transition HGF**

Using Bayesian inference approximated by Markov Chain Monte Carlo (MCMC) methods, the complete task dataset was fit to three different learning models of which the categorical state-transition HGF performed best on model comparisons (Supplement). The categorical state-transition HGF learns transition probabilities between observed discrete categories using variational Bayesian inference in a predictive coding-like manner (Fig. 3a, Methods). Here, the categories tracked by

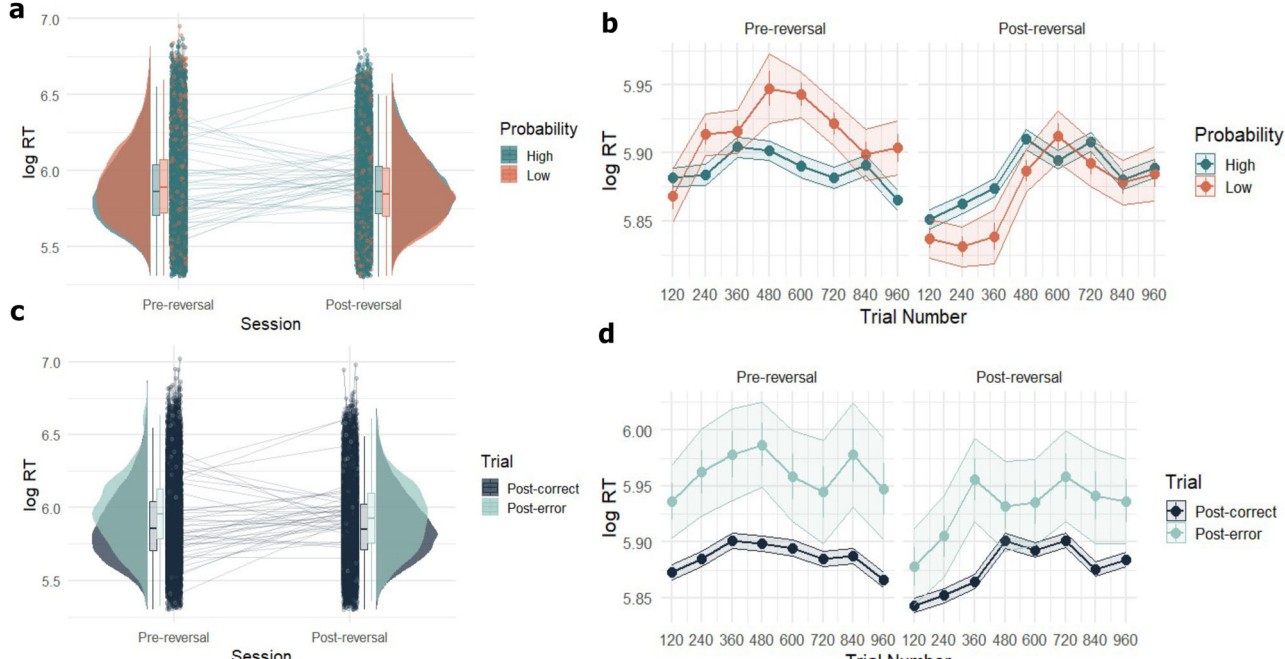

**Fig. 2 | Model-agnostic behavioural results. a** Distributions (boxplots and raincloud plots) of mean log-transformed RT for correct responses grouped according to High Probability (green) and Low Probability (orange) trials across Pre- and Post-reversal sessions for *n* = 42 participants. In the boxplots, the central line indicates the median, the bounds of the box represent the 25th and 75th percentiles (interquartile range, IQR), and the whiskers extend to the minima and maxima within 1.5 × IQR. Dots show individual participant means, and connecting lines track participants across sessions. **b** Mean log-transformed RT across time for High Probability (green) vs Low Probability (orange) trials across Pre- and Post-reversal sessions for *n* = 42 participants. Data points indicate the mean for each block (120 trials), and bars indicate the SEM. **c** Distributions (boxplots and raincloud plots) of mean log-

transformed RT for all responses grouped according to whether trials followed an error (post-error, in turquoise) vs a correct (post-correct, in dark blue) response for *n* = 42 participants. In the boxplots, the central line indicates the median, the bounds of the box represent the 25th and 75th percentiles (IQR), and the whiskers extend to the minima and maxima within 1.5 × IQR. Dots show individual participant means, and connecting lines track participants across sessions. **d** Mean log-transformed RT across time for Post-error vs Post-correct trials across Pre- and Post-reversal sessions for *n* = 42 participants. Data points indicate the mean for each block (120 trials), and bars indicate the SEM. Source data for all panels are provided as a Source Data file.

the HGF represent the four possible stimulus locations on the SRT, resulting in a total of 16 possible transitions. The HGF is implemented as a network of nodes, each representing a probabilistic belief distribution about some part of the task: the received sensory observation *u* of category *j*, the transition from the previous category *i* that was observed $c_{i,j}$, the probability of that transition occurring on the next trial $b_{i,j}$, and the in-time change of the probability in log-odds $x_{i,j}$. On each trial, the observed category is compared to its prediction to produce a measure of surprise $\Im$ (Eq. 1) and a prediction error $\delta$ (Supplement). These quantities are passed up through the hierarchy of nodes and used to update beliefs about transition probabilities, which in turn supply predictions for the following trial (Methods, Supplement).

The HGF holds Gaussian belief distributions about $x_{i,j}$, with $\mu$ representing the mean of the current belief distribution about the transition probability and the precision $\pi$ representing the certainty in that belief distribution (see Fig. 3b, c for an example of how these two values evolve over time for a specific transition probability). Belief updates are larger in magnitude when prediction errors are large, and when the precision $\pi$ of the current belief distribution is low. At each trial, the means $\mu$ of the belief distribution are transformed to probability space and normalised to construct the final expected transition probabilities (Fig. 3d displays an example session where these 16 transition probabilities are learnt over time). The learning of transition probabilities is contingent on the parameter $\omega$, which represents the expectation of general volatility in the environment (Methods, Eq. 3). Higher values of $\omega$ lead to less certainty and faster updating of beliefs.

To connect the belief dynamics of the HGF to the log-transformed reaction times, I (Eq. 1), $U_{unexpected}$ (Eq. 2), $U_{unexpected}$ Eq. 4), *Post − error*, and *Post − reversal* were included as predictors in a response model (Eq. 5). Thus the full model has 8 free parameters: the expectation or belief of general environmental volatility $\omega$, the 6 beta parameters (including intercept term) of the response model, and the standard error of the regression $\sigma$.

At the group level, the mean(SD) estimates for each regressor of the response model were as follows: $\beta_0 = 5.83(0.68)$, $\beta_1.\Im = 0.05(0.25)$, $\beta_2.U_{expected} = −0.02(0.89)$, $\beta_3.U_{unexpected} = −0.15(1.5)$, $\beta_4.Post − error = 0.10(0.07)$, and $\beta_5.Post − reversal = −0.01(0.05)$ (Fig. 4a). To test if there were consistent effects across the whole group, we conducted one-sample *t*-tests separately for each beta estimate. We found that only $\beta_4.Post − error$ had a significant effect ($t(41) = 10.73$, $p = 1.73 \times 10^{-13}$, Cohen's $d = 1.67$, 95% CI [0.088, 0.128]), indicating that the post-error slowing effect was most stably present across the group. However, at the individual-level, we see clear indication of inter-individual variation in the distributions of estimates for each regressor of the response model (Fig. 4 b-h).

## M1 Glx levels predict computational indices of uncertainty

After completing the behavioural task, participants completed a separate MRI session during which spectroscopy data were collected (Methods). We fit an LME model on log RT with task session (Pre- vs Post-reversal) and metabolite levels as regressors (LME Model 4). We found a significant interaction between M1 Glx and session

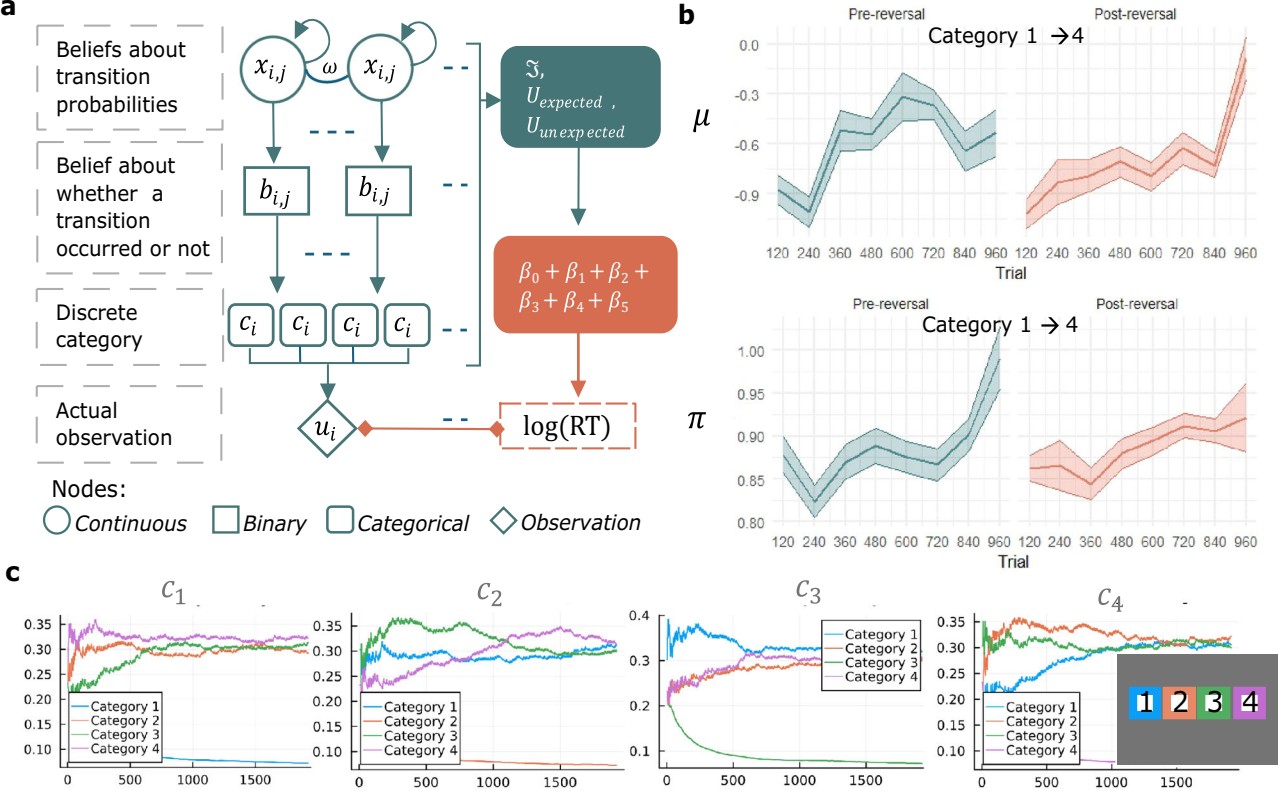

**Fig. 3 | Categorical state-transition Hierarchical Gaussian Filter (HGF). a** Schematic description of the categorical state-transition HGF. Beliefs are represented at probabilistic nodes, with higher-level continuous nodes ($x_{i,j}$) predicting the changing transition probabilities represented by lower-level nodes ($b_{i,j}$), contingent on the belief about general environmental volatility ($\omega$) that is shared between all $x_{i,j}$. Transition probabilities are normalised to proper probability distributions in categorical nodes $c_i$. Straight arrows indicate dependencies in the HGF's generative model, and curved arrows indicate a dependency on the node's own previous time step. Dotted lines indicate the multiplicity of nodes. In our implementation, there are 16 higher-order nodes, one for each possible transition between four categories (4×4). Key belief states from the HGF are mapped to reaction times by means of a response model (indicated in orange). **b** Evolution of the higher-level belief distributions $\mu$ (top graph) about transition probabilities and the corresponding posterior precision $\pi$ (bottom graph) at the continuous node ($x_{i,j}$) for a specific

transition (namely, from Category 1 at trial t-1 to Category 4 at trial t), at an example session. Each data point corresponds to the mean of the state transition belief distribution ($\mu$) or posterior precision ($\pi$) in bins of 120 trials (x-axis), across each session of the task (Pre-reversal in green, and Post-reversal in orange). Lines indicate mean values, and shaded areas represent ± SEM. Source data are provided as a Source Data file. **c** The HGF's inferred state-transition probabilities over time in the same example session. Each panel represents transitions from one of the 4 discrete categories (i.e., possible stimulus locations). Transition beliefs (y-axis) across time (x-axis) are estimated for each possible transition. For example, the leftmost panel (titled c1) represents the transitions from Category 1 (blue) to: Category 1 (blue), Category 2 (orange), Category 3 (green), and Category 4 (purple). As the task does not contain transitions from Category 1 to Category 1, the estimate of this transition quickly approaches 0.

($t(64,470) = -3.17$, $p = 0.001$, $b = -0.065$, $SE = 0.021$, 95% CI [−0.105, −0.025]), suggesting that M1 Glx may affect reversal learning in particular.

Next, we conducted separate Pearson's correlations between the computational model estimates and metabolite levels. We examined prediction errors ($\delta$) at the highest continuous level of the HGF hierarchy; specifically, the top-level node tracking beliefs about transition probabilities, which reflects the magnitude of belief updating in response to unexpected changes in transition probabilities. To calculate each participant's mean prediction error ($\delta$), high-level $\delta$ corresponding to the 16 possible transitions were averaged for each participant and included in the correlation. This yielded a significant positive correlation ($r(37) = 0.37$, $p = 0.02$, 95% CI [0.04,0.63]). In other words, higher levels of M1 Glx were associated with larger prediction errors ($\delta$). To rule out task performance-related confounds, we confirmed that M1 Glx was not significantly correlated with task accuracy (Supplement).

In contrast, we found a significant negative correlation between M1 Glx levels and participants' mean belief about volatility ($\omega$) ($r(37) = -0.40$, $p = 0.018$, 95% CI [−0.65, −0.08]), indicating that

higher Glx levels are associated with lower beliefs about volatility. Notably, in our implementation of the HGF, $\omega$ is the sole free parameter and governs belief updating at higher levels of the hierarchy. As such, variation in $\omega$ shapes both volatility estimates and their influence on the magnitude of prediction errors ($\delta$). Indeed, a separate supplementary correlation analysis confirmed a strong inverse relationship between these quantities (Supplement), indicating that participants with lower inferred volatility exhibited larger high-level prediction errors. Taken together, these findings suggest that the above associations with Glx may reflect its influence on a shared neurocomputational mechanism underlying both volatility beliefs and high-level prediction errors.

Finally, to confirm the regional- and metabolite- specificity of our findings, we ran supplemental control analyses on M1 GABA and a control voxel placed on the occipital cortex. We found no relationships between M1 GABA and behaviour, confirming that these findings are specific to Glx (Supplement). In addition, we examined Glx levels from MRS data acquired from a control voxel; this yielded no significant results, confirming the M1 regional-specificity of our findings (Supplement).

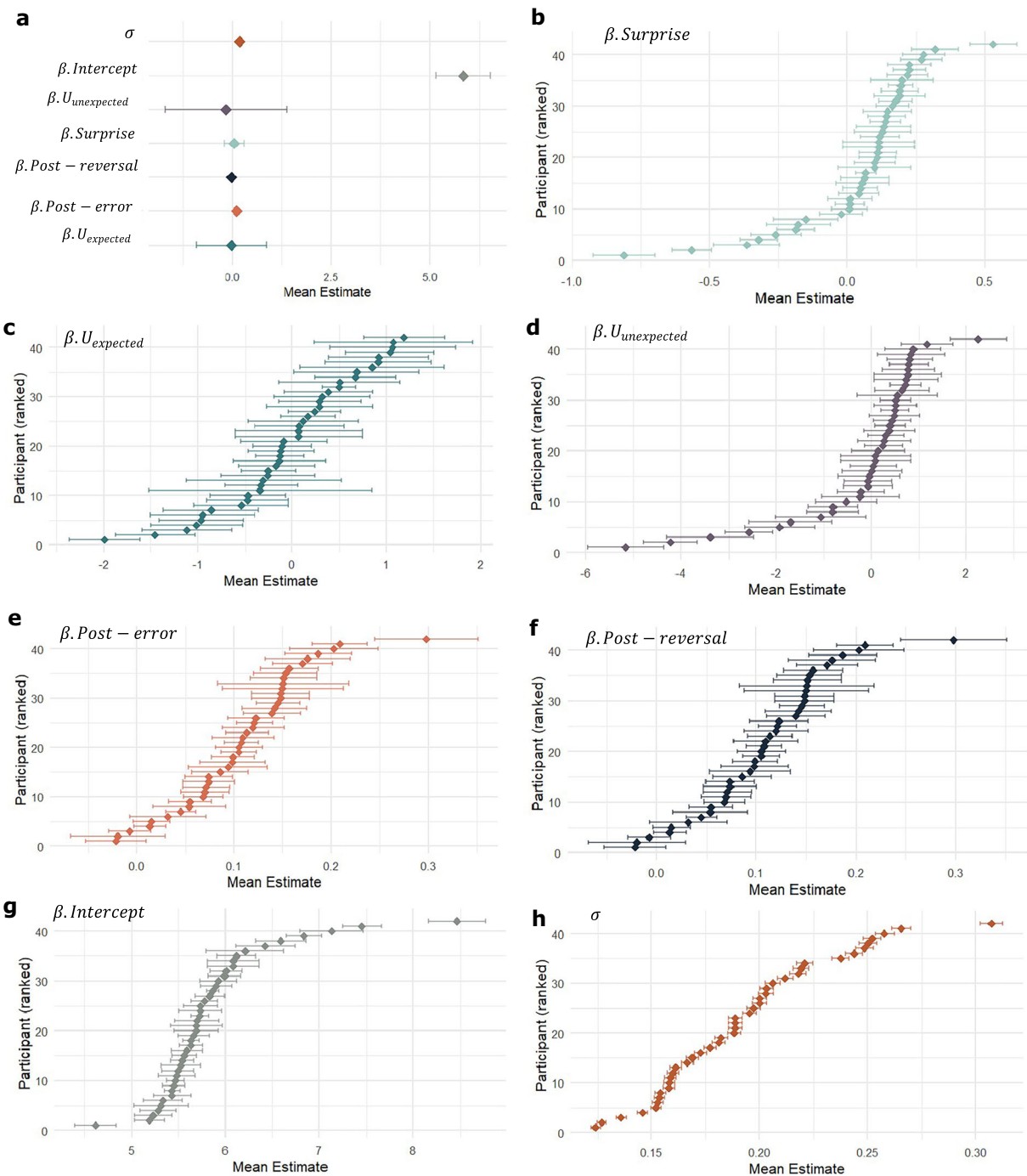

**Fig. 4 | Categorical state-transition HGF results. a** Mean beta weights at the group level for each of the predictors included in the response model. The HGF response model maps beliefs estimated from the perceptual model and relevant task regressors onto participants' log-transformed RT by means of a linear regression. Data points indicate the mean estimates (x-axis), with bars representing SEM. All the predictors (y-axis) were included in the same response model. **b–h** Individual participant-level mean beta estimates for response model predictors $\beta$.Surprise (pale green), $\beta$. (dark green), $\beta.U_{unexpected}$ (purple), $\beta.Post - error$ (orange), $\beta.Post - reversal$ (dark blue), $\beta.Intercept$ (grey), and standard error of the regression ($\sigma$) (brown), respectively. The x-axis indicates the mean estimate, and the y-axis indicates each participant arranged in ascending order of the value of their individual mean estimate. Data are from $n = 42$ independent participants. Each participant provided a single dataset of behavioural responses, from which regression parameters were estimated. Groups compared are different response model predictors; no separate control group was used, as all predictors were jointly estimated within the same model. Each data point corresponds to the participant's mean, with error bars representing SEM. Posteriors were estimated through Monte Carlo Markov Chain (MCMC) inference using 4 chains and 2000 iterations corresponding to each chain. Source data for all panels are provided as a Source Data file.

## Relationship with Trait Anxiety

We implemented an LME model on log-transformed RT with task session and Spielberger State-Trait Inventory (STAI)[51] trait anxiety scores as predictors, and STAI state anxiety as a controlling variable (LME Model 5). This yielded a significant interaction between trait anxiety and task session ($t(79,200) = 15.27$, $p = 2 \times 10^{-16}$, $b = 0.002$, $SE = 0.0001$, 95% CI [0.002, 0.002]). In other words, individuals high in trait anxiety, although overall slower compared to those with low trait

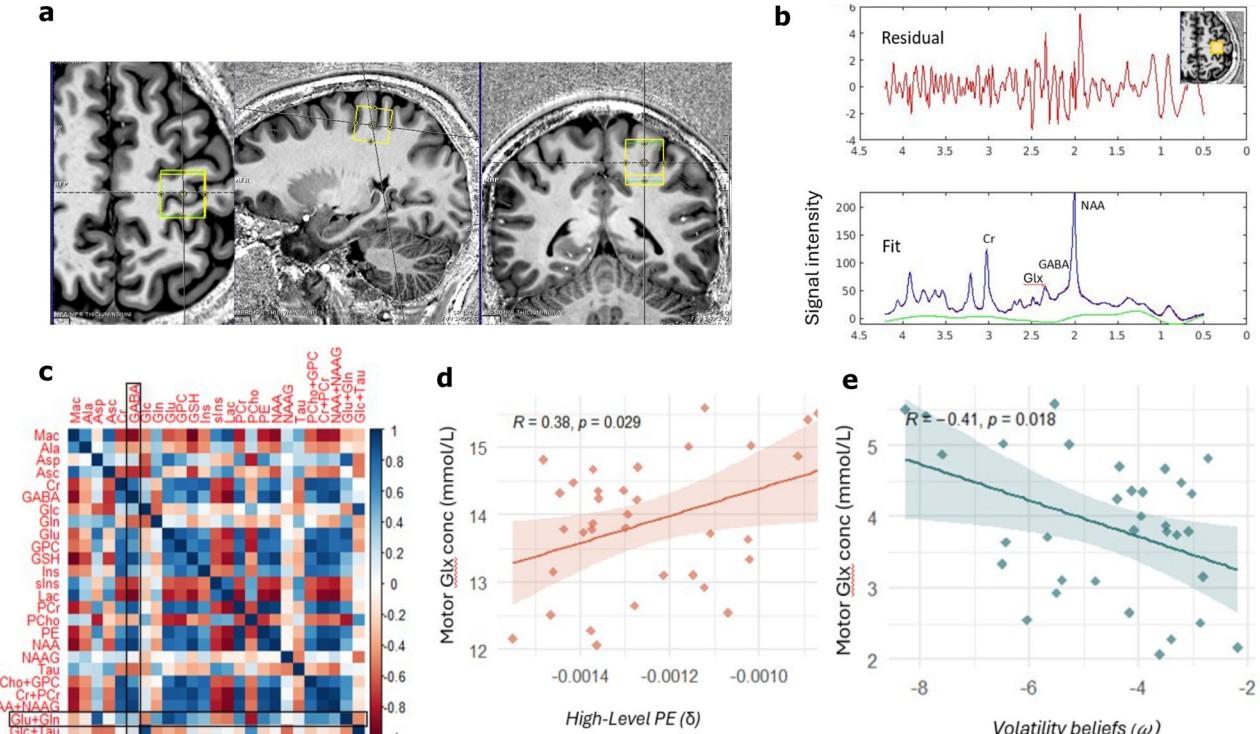

**Fig. 5 | MRS spectroscopy. a** Motor cortex (M1) voxel placement. **b** Example MR spectra for a sample participant. Upper panel: raw spectra containing the residuals after LC Model fitting; Lower panel: LC Model fit with peaks of all metabolites. The main Glx (Glu + Gln) peak resonates between 2.2 and 2.4 ppm, while GABA shows peaks at approximately 3.0ppm, 1.9ppm and 2.2 ppm. **c** Correlation matrix of all the quantified metabolites (uncorrected) in M1 voxel across *n* = 37 participants. The metabolites GABA and Glx (Glu + Gln) are outlined. The colour bar indicates the strength of the correlation, with dark blue indicating positive correlation and red indicating negative correlation. **d** Significant positive correlation between M1 Glx levels and high-level prediction errors (Pearson's correlation, r = 0.38, *p* = 0.02, 95%

CI [0.04,0.63] two-sided, uncorrected) for *n* = 37 participants. PE = prediction errors, denoted by the δ symbol. The regression line represents the best-fitting linear model, and the shaded area indicates ± SEM around the fitted line. Each point corresponds to an individual participant. **e** Significant negative correlation between M1 Glx levels and predicted belief about volatility (Pearson's correlation, r = −0.40, *p* = 0.018, 95% CI[−0.65, −0.08], two-sided, uncorrected) for *n* = 37 participants. Volatility beliefs are denoted by the ω symbol. The regression line represents the best-fitting linear model, and the shaded area indicates ± SEM around the fitted line. Each point corresponds to an individual participant. Source data for **c**–**e** are provided as a Source Data file.

anxiety, were found to be faster after the reversal. To visualise this effect, we created High and Low Trait Anxiety groups based on a median split of the trait anxiety scores (Fig. 6a).

Next, to corroborate the above model-agnostic findings, we examined the relationship between trait anxiety (as calculated by STAI-Trait scores, Methods) and the HGF response model beta estimate for task session $\beta$. *Post − reversal*. This yielded a significant negative correlation ($r(42) = −0.51$, $p = 5 \times 10^{-4}$, 95% CI [−0.70, −0.24]), suggesting that the magnitude of the relationship between response speed and the reversal may be predicted by trait anxiety scores (Fig. 6b). Contrary to our hypothesis, we found no relationships between beliefs about volatility ($\omega$) and trait anxiety ($r(42) = −0.17$, $p = 0.29$, 95% CI [−0.44 0.14]).

## Discussion

This study provides evidence of motor glutamate + glutamine (Glx) contribution to probabilistic learning in humans, specifically in environments that involve changing or volatile statistics. The hierarchical Bayesian model we implemented offers a robust framework for understanding how individuals build and update beliefs about the world, capturing both the explicit and implicit processes involved in learning under uncertainty and the complex belief trajectories of non-binary response options. Our findings suggest that variations in M1 Glx may serve as a neural signature of individual differences in adaptive learning under uncertainty.

We found that participants implicitly learned the probabilistic structure of a four-choice sensorimotor task and adapted to the probabilistic reversal. Evidence of post-error slowing suggested that participants adjusted their behaviour following errors to improve subsequent performance. At the group-level, post-error slowing most consistently contributed to reaction times during probabilistic reversal learning, corroborating the model-free results. Although response times are often assumed to reflect internal uncertainty estimates, our HGF response model indicated that reaction times were not consistently modulated by belief-based predictors such as expected or unexpected uncertainty or surprise. Apart from a reliable post-error slowing effect, the belief-related regressors showed no significant group-level effects, suggesting that reaction times may capture the behavioural consequences of learning, such as post-error slowing, rather than belief updating itself. However, at the participant-level, we see considerable inter-individual differences in the response model beta estimates, including surprise, expected and unexpected uncertainty, and post-reversal learning, revealing meaningful evidence of individual variability in uncertainty-related computations (Fig. 4).

While the HGF model used here does not include an explicit learning rate parameter, learning rates are dynamically governed by participants' inferred volatility beliefs, which shape the rate at which beliefs are updated. This dynamic updating mechanism effectively serves as an implicit, trial-by-trial learning rate. Consistent with prior work employing the HGF framework[12,27,52,53], our approach uses a separate regression model to relate these computational variables to behaviour. Although less mechanistic than

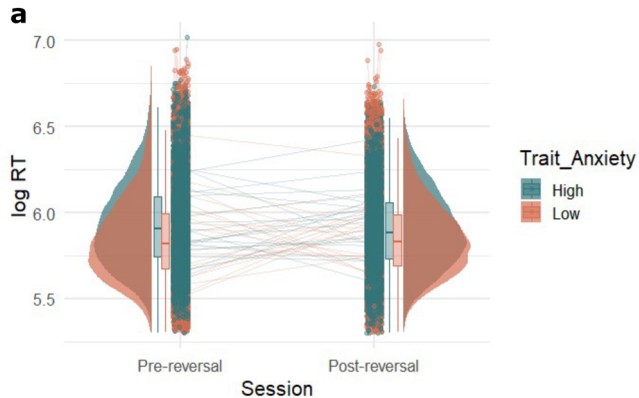

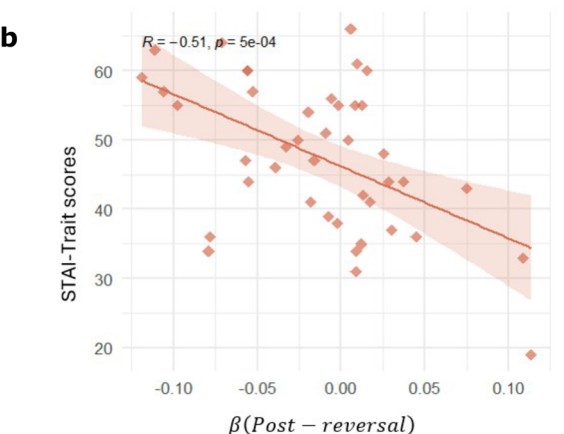

**Fig. 6 | Relationship with Trait Anxiety. a** Distributions (boxplots and raincloud plots) of mean log-transformed RT grouped according to High Anxiety (green) versus Low Anxiety (orange) across Pre- and Post-reversal sessions for n = 42 participants. Groups compared are defined by trait anxiety levels. In the boxplots, the central line indicates the median, the bounds of the box represent the 25th and 75th percentiles (IQR), and the whiskers extend to the minima and maxima within 1.5 × IQR. Dots show individual participant means, and connecting lines track participants across sessions. Anxiety levels were split by median trait anxiety scores for visualisation only and analyses were focused on continuous trait anxiety scores. **b** Significant negative correlation between HGF response model beta estimate for post-reversal and trait anxiety scores for *n* = 42 participants (Pearson's correlation, r = −0.51, *p* = 5 × 10⁻⁴, 95% CI [−0.70, −0.24], two-sided, uncorrected). Trait anxiety was measured using the trait anxiety subscale of the Spielberger Trait-State Anxiety Inventory (STAI, see Methods). The regression line represents the best-fitting linear model, and the shaded area indicates ± SEM around the fitted line. Each point corresponds to an individual participant. Source data are provided as a Source Data file.

fully generative models, this allows us to test targeted hypotheses about how specific computational signals and task-related parameters influence response time dynamics. Future work could extend these insights by incorporating more mechanistic response models.

We further linked the model-free and model-based results by means of a supplemental analysis where computationally-estimated surprise was matched against the trial-by-trial behavioural data (Supplement). We found that the trial-by-trial evolution of surprise aligned with our model-free behavioural findings, with surprise decreasing as the task progressed. At the same time, there was no significant interaction between surprise and the post-reversal session, suggesting that participants either adapted to the reversal or that any effects diminished over time. These results add to the growing body of literature involving different populations, including recent work disentangling

volatility from stochasticity[54], demonstrating that volatility enhances—rather than impedes—learning[5,10,12,14,27,52].

Although more specific constructs, such as the Intolerance of Uncertainty Scale[55], may better capture sensitivity to uncertainty, we focused on trait anxiety due to its broader clinical relevance and well-documented links to uncertainty processing. Based on prior findings linking anxiety and related conditions to misestimation of uncertainty and overestimation of volatility[9,14,19,22,27], we had hypothesised a relationship between trait anxiety levels and beliefs about volatility (ω). In contrast, we found no association between trait anxiety and volatility beliefs (ω) in our task, suggesting that individuals with high trait anxiety may not be as negatively affected by uncertainty in implicit probabilistic learning tasks that lack valence-related elements (such as reward, punishment, or threat). Notably, although individuals with high trait anxiety were generally slower or more cautious, they responded more quickly after the reversal, suggesting that they effectively adapted to the changed contingencies. This is in line with recent findings that individuals high in trait anxiety may be faster to update their expectations following reversals[21]. Meanwhile, recent large-scale studies have questioned reports of altered uncertainty processing in anxiety[29,30]. To contribute to this ongoing discussion, we conducted exploratory analyses examining relationships between trait anxiety and model-derived uncertainty estimates (volatility, expected uncertainty, and unexpected uncertainty), none of which yielded significant correlations (Supplement). This suggests that the relationship between trait anxiety and belief-based uncertainty may be weaker or more context-dependent than previously assumed.

Our key contribution lies in identifying a neurochemical correlate of inter-individual differences in uncertainty processing, specifically within the context of a non-binary, probabilistic learning task. We introduce a categorical state-transition extension of the HGF that captures belief updating over multiple discrete outcomes. This model allows us to formally characterise how participants infer transition probabilities in a four-choice environment, moving beyond traditional binary formulations. We found a significant interaction between M1 Glx and task session, suggesting that Glx in the primary motor cortex plays a role in reversal learning in a probabilistic sensorimotor task. Numerous studies on non-human primates and rodents have pinpointed the neural basis of reversal learning to glutamatergic mechanisms[46,47,56–61]. We establish a similar link in humans and delve deeper into this by means of computational modelling.

Learning is driven by prediction errors - the difference between what was expected and what actually occurred[62,63]. Our model-based findings suggest that higher M1 Glx levels are associated with greater prediction errors, reinforcing the idea that glutamate-related plasticity supports error-driven learning. Importantly, in hierarchical models, the average prediction errors, particularly at higher levels, can also be interpreted in terms of information gain, capturing the extent to which surprising outcomes prompt updates to more abstract beliefs about environmental structure. In this framework, larger high-level prediction errors reflect a greater need to revise internal models, suggesting that elevated Glx may support both associative learning and broader belief updating under uncertainty.

Our model captures individual differences in inferred environmental volatility, offering insight into how participants internally model the stability or instability of the probabilistic structure, even in the absence of explicit volatility manipulations. Notably, we also observed a negative correlation between Glx levels and beliefs about volatility, indicating that participants with higher M1 Glx levels might have lower expectations of general volatility in the environment. This effect is particularly relevant given that volatility beliefs modulate the learning rate by determining how much weight is assigned to new information. Furthermore, a supplementary correlation analysis revealed a strong negative relationship between high-level prediction errors and volatility beliefs (Supplement). In other words, when the

environment was perceived as more stable, surprising outcomes had greater computational impact, resulting in larger belief updates. This relationship provides a mechanistic bridge between Glx levels, volatility beliefs, and prediction error dynamics: elevated Glx may lead individuals to expect a more stable environment, thereby amplifying the impact of unexpected outcomes and enhancing the learning signal when predictions are violated. In contrast, we found no significant relationships between M1 GABA and uncertainty processing, confirming the glutamatergic specificity of these findings. The absence of a relationship between GABA and probabilistic learning in our study suggests that the excitatory actions of Glx, rather than inhibitory regulation via GABA, are more central to the dynamics of learning in this task.

While the present study provides valuable insights into the role of Glx in learning under uncertainty, there are a number of limitations that should be considered. First, baseline MRS measurements may be considered only indirectly indicative of neurotransmitter signalling. Second, as our MRS analyses were focused on Glx, which contains signals from both glutamate and glutamine, we cannot attribute our findings to glutamate in particular. Glutamine is a precursor to glutamate and the two are enzymatically linked[64]. Although 7T in principle offers improved spectral resolution, we chose to focus on Glx, as our data showed notable quantification uncertainty for glutamate and glutamine when estimated separately, and the two metabolites were weakly correlated. Specifically, this uncertainty was indexed by elevated Cramér-Rao lower bounds (CRLB), indicating less reliable estimates when attempting to separate the metabolites. This is consistent with recent work emphasising the challenges of reliably separating these metabolites at 7 T, which depend heavily on the choice of sequence, voxel placement, and the brain region studied[65]. Focusing on Glx provides a more robust and conservative estimate of excitatory neurotransmitter levels, reducing the risk of overinterpreting potentially noisy or overlapping signals. Third, the post-reversal effects on behaviour and belief updating cannot be conclusively attributed solely to the reversal itself, as they may also reflect general effects of time or task progression. Future work with matched control conditions or counterbalanced designs may help to disambiguate these effects.

Prior studies of uncertainty processing have implicated frontal and subcortical regions such as the orbitofrontal cortex, anterior cingulate cortex, and striatum, particularly in value-based or explicit outcome monitoring paradigms[5,8,10] In contrast, our study employed an implicit learning paradigm in which participants were neither consciously aware of the underlying probabilistic structure nor required to explicitly make choices or respond to uncertainty. Given this design, we hypothesised that uncertainty is processed in a more automatic, sensory fashion, with sensorimotor regions tracking environmental regularities through accumulated experience. The primary motor cortex (M1) is a well-suited MRS target given its established role in implicit motor learning and sensorimotor plasticity. Prior work has shown that M1 exhibits neurochemical changes, such as GABA modulation, in response to learning demands even in the absence of explicit awareness[66]. Moreover, M1 has been implicated in tracking environmental regularities and processing feedback-related and sensory uncertainty[67–69], suggesting it may play a role in implicit learning under uncertainty.

We were able to confirm the regional specificity of our findings based on the results of a control analysis conducted on an occipital cortex voxel. However, we note that other brain regions and neurotransmitter systems likely play significant roles in learning and decision-making under uncertainty. Future studies incorporating multi-region MRS imaging or other neuroimaging techniques (such as fMRI) could provide a more comprehensive understanding of the neurobiological mechanisms underlying uncertainty processing in humans. Additionally, examining excitatory mechanisms in clinical populations with anxiety or other psychiatric conditions could provide

important insights into how neurochemical imbalances contribute to maladaptive learning and decision-making.

In conclusion, this study presents an investigation into the neural basis of probabilistic learning under uncertainty, revealing M1 Glx as an important neurochemical correlate of volatility beliefs and prediction errors during implicit learning. We introduce a model of probabilistic learning – the categorical state-transition HGF – that captures the complexity of non-binary choices in the classic four-choice probabilistic Serial Reaction Time task. By linking computational estimates of uncertainty processing with neurochemical data, we provide a deeper understanding of inter-individual differences in adaptive behaviour in unpredictable environments. These results also offer potential avenues for future research into how neurochemical variations in M1 Glx may influence learning and decision-making in both healthy and clinical populations.

## Methods
### Participants
43 right-handed participants (20 Female: 23 Male), aged 19-39 years (*Mean* = 28.37, *SD* = 4.75) participated in the study. Right-handed participants with normal or corrected-to-normal vision were recruited through social media, local classified advertisements, and University mailing lists. Participants reported both sex assigned at birth and gender, which were collected to ensure a sex-balanced sample. Sex/gender was not analysed as a variable of interest or included as a covariate. Participants were paid £40 for their time. The sample size was based on a power calculation using effect sizes from recent literature[66,70–72]. To estimate a correlation of $r = 0.4$, a sample size of n = 42 (*power* = 0.8, *Type I error* = 0.05) was required to observe our primary effect of interest, namely a relationship between the metabolite of interest and behaviour.

### Procedure
This study was approved by and conducted in accordance with the regulations of the University of Cambridge Psychology Research Ethics Committee (PRE.2020.127). Written informed consent was obtained from all participants. The study was completed over two sessions.

In the first session, participants completed the probabilistic SRT task[3,73] on a desktop computer at a viewing distance of 50 cm from the screen in a darkened room. Stimuli were presented using Psychtoolbox version 3[74] in MATLAB R2019b. Stimuli were displayed on a 24" monitor at a resolution of 1920 x 1080. Each trial consisted of the appearance of a cross in one of the 4 squares (Fig. 4a). Participants were instructed to indicate the location of the cross by pressing a key corresponding to the four different possible locations. Stimulus duration lasted 3000 ms with an inter-stimulus interval of 200 ms. A beep was played to alert participants to incorrect or missed responses. Following the first session, participants commenced the second session after a 15 minute break. The probabilistic version of the SRT task differs from the deterministic version in its use of second-order conditional sequences[75,76]. Here, unbeknownst to the participants, the position of the stimulus is determined by one of two Markov-chain sequences (Fig. 1c):

Sequence A: 1-2-1-4-3-2-4-1-3-4-2-3-1-2-1-4-3-2-4-1-3-4-2-3
Sequence B: 1-3-2-3-4-1-2-4-3-1-4-2-1-3-2-3-4-1-2-4-3-1-4-2

The position on trial t is determined by the position of the two previous trials, t-1 and t-2. The two deciding sequences differ in the second-order conditional occurrence, with one occurring 85% of the time and the other 15%[76–78] (Fig. 1b–d). This means that, although transitions are derived from two predefined sequences (Sequence A and Sequence B), both high and low probability transitions occur within the same unified probabilistic framework. In other words, these transitions do not reflect switches between entire sequences but instead represent more or less frequent transitions within the same overarching Markov structure governing the task.

In the first session, there is an 85% probability of trial t-1 and t-2 leading to trial t in Sequence A, and a 15% probability of transitioning to trial t in Sequence B. Participants are hypothesised to develop expectations of upcoming targets in the probable sequences by implicitly learning this second-order conditional information. Following the completion of the first session, the sequences were reversed, making Sequence B the more probable sequence and Sequence A the less probable sequence.

In the same session, participants also completed the Spielberger State-Trait Inventory (STAI), a 40-item questionnaire to assess individuals' anxiety across state and trait domains[51].

In the next session, participants completed a 7 T MRI scan at the Wolfson Brain Imaging Centre, University of Cambridge. The scan took place on a separate day from the behavioural testing session but was conducted within the same week. A Siemens 7T Magnetom Terra scanner (Siemens, Erlangen, Germany) with a single-channel transmit and 32-channel receiver head coil (Nova Medical, Carson, CA) was used. T1-weighted MP2RAGE structural scans (repetition time $TR$ = 4300 ms, echo time $TE$ = 1.99 ms, bandwidth= 250 Hz/px, voxel size = 0.75 mm$^3$, isotropic field of view = 240 x 240 x 157 mm, acceleration factor = 3, flipangle = 5/6° and inversion times $TI$ = 840/2370 ms) were acquired. Spectra were acquired using a short-echo semi-LASER sequence[79,80] with repetition time $TR$ /echo time $TE$ = 5000/26 ms, 64 repetitions. Pre-scan optimisation included FASTESTMAP $B_0$-shimming[81], unsuppressed water-peak series $B_1$ calibration, and VAPOR water suppression calibration[82]. MR spectra were obtained from a 2x2x2 cm$^3$ voxel of interest (VOI) placed on the primary motor cortex (M1) while participants were at rest. The voxel placement was manually centred on the omega- or epsilon-shaped left hand knob area using the central sulcus as a landmark[66,83] (Fig. 5a). The VOI was placed to capture the entire hand knob and to exclude the dura. For the motor VOI, the anatomical landmarks could be most clearly observed from the coronal view (Fig. 5a). To confirm the regional specificity of the findings, we also obtained MR spectra from a control voxel placed in the occipital cortex, acquired with identical parameters (Supplement).

## Analyses

Data were cleaned and analysed using R 4.3.3. Behavioural data outliers were identified as RT < 200 ms, while slow RT outliers were computed separately for each participant as trials with RT more than 2 standard deviations from their overall RT. These exclusions were applied to minimise the influence of extreme values likely reflecting attentional lapses, momentary disengagement, or other non-task-related factors. We employed linear mixed-effects models (LME) to examine the effect of task, questionnaire, or neural variables on log-transformed RT using R packages lme4[84] and lmerTest[85]. This approach allows us to account for both fixed effects (variables that are consistent across all participants) and random effects (variables that vary across participants). Each participant was included as a random intercept in the LME models. LME Model 2 included random slopes for trial number by participant, allowing us to account for individual differences in learning trajectories across trials. The inclusion of random slopes was based on a model comparison procedure, which showed that this model fit the data significantly better than a model without random slopes (Supplement). LME models testing for effects of stimulus probability (LME Models 1 and 2, Results, Fig. 2) on response time were focused on correct responses only. The following LME models were employed:

$$log(RT) \sim Stimulus\ probability + Session + (1|\ Participant)$$

LME model 1

$$log(RT) \sim (Stimulus\ probability + Session) * Trial\ Number \\ + (1 + Trial\ Number|Participant)$$

LME model 2

$$log(RT) \sim Session * Trial\ type + (1|Participant)$$

LME model 3

$$log(RT) \sim Session * Metabolite\ concentration + (1|Participant)$$

LME model 4

$$log(RT) \sim Trait\ Anxiety * Session + (1|Participant) + State\ Anxiety$$

LME model 5

where stimulus probability refers to high vs low probability trials, session refers to pre- vs post-reversal task sessions, trial type refers to post-error vs post-correct trials, trial number is used as a continuous measure of time, Trait Anxiety indicates STAI-Trait subdomain scores, and State Anxiety indicates STAI-State subdomain scores.

The acquired MR spectra were corrected for eddy current effects, and phase and frequency drifts using MRspa (www.cmrr.umn.edu/downloads/mrspa/), in line with the current best-practice recommendations for pre-processing[86]. Metabolites between 0.5 and 4.2 ppm – including Glx and GABA- were quantified using LCModel v6.31[87]. Metabolite concentrations are reported with reference to the water signal (sometimes referred to as the "absolute" concentration)[88]. They were corrected for inter-individual differences in tissue volume. Details of the tissue segmentation and correction procedure can be found in the Supplement. The following exclusion criteria were used for quality control: water linewidth > 15 Hz, signal-to-noise ratio <40, and Cramér-Rao lower bounds > 2 standard deviations from the group median[89–91]. Following the quality control exclusions, data from 37 participants remained for the M1 voxel. Outliers were identified as those above or below the cutoff of 2 standard deviations from the group median.

Specific relationships between the computational modelling estimates and metabolite concentrations or questionnaire measures were assessed by means of Pearson's correlations. All statistical analyses were two-tailed unless otherwise stated.

## Computational cognitive modelling

We modelled behaviour using a categorical state-transition HGF. The HGF is a popular model of perception and learning in volatile environments[6,35] that has recently been generalised to a flexible network formalisation[36]. It is a variational Bayesian predictive processing model, in which perception is modelled as variational Bayesian inference on otherwise inaccessible environmental states. This inference relies on a generative model of how the environment changes and produces sensory observations, which is then inverted through Bayesian inference to estimate the current environmental state given incoming sensory input. In the generalised HGF, the generative model is implemented as a network of nodes representing different states of the environment. States can either be binary, categorical, or continuous, and can connect with value couplings, where the expected value of a node depends on another, or with volatility couplings, where the noisiness of each node depends on another. Each node type has its own set of update equations and free parameters, dependent on how they are connected. A detailed formal introduction can be found in the original manuscript[36]. Here, we present the updates relevant specific to our study.

We built upon the generalised HGF to create a categorical state-transition HGF, which specifically learns the probabilities of different transitions between categorical states. It consists of the following nodes: a categorical input node $u$ which registers the categorical input in the task (i.e., the stimulus location); a set of categorical state nodes $c_i$ that store the type of transition made, and the normalised predicted transition probabilities; a set of binary nodes $b_{i,j}$ for each transition

type that stores whether that transition occurred, and the predicted probability of it occurring in the future; and a set of continuous nodes $x_{i,j}$ that track the changing (log)probabilities of each of those state-transitions. There are 16 higher-order continuous nodes, each representing a possible transition between the four categories. First, node belief distributions are updated based on prediction errors. Since there is no observational uncertainty, $c_i$ is simply updated by setting the belief about the observed category to be equal to the actual category. Similarly, nodes $b_{i,j}$ are set to 1 when the transition occurred, and 0 when it did not. Nodes $x_{i,j}$ are updated as continuous value parents to binary nodes; these update equations have been reported in the Supplement.

For the response model of the HGF, we follow the approach used by Marshall et al. [52], and implement a linear regression linking various belief states of the HGF to the log-reaction times of participant's responses.

First, the surprise ($\Im$) or unexpectedness of the observed transition at each trial $t$ based on the agent's internal model is calculated as the negative log probability of the transition, under the model's prediction (See supplement for prediction calculation):

$$\Im_t = -\log\big(\mathrm{pdf}\big(\mathrm{Cat}(\hat{\mu}_{ci}), u^t\big)\big) \tag{1}$$

where $\hat{\mu}$ is the mean prediction, $c_i$ represents the categorical state node that stores the type of transition made, $\hat{\mu}_{c_i}$ the categorical probability distribution, and $u^t$ represents the node containing the actual observations at time $t$.

In addition, we quantified expected and unexpected uncertainty[9]. The expected uncertainty is the uncertainty regarding whether the observed transition would have occurred, multiplied with the precision of the belief at $x_{i\setminus j}$:

$$U_{expected} = \mathrm{S}(\hat{\mu}_{x_{i,j}})\Big(1 - \mathrm{S}(\hat{\mu}_{x_{i,j}})\Big) \cdot \frac{1}{\pi_{x_{i,j}}} \tag{2}$$

where $S(\hat{\mu}_{x_{i,j}})$ is the logistic transformation of the mean prediction for the node tracking the probability of the observed transition.

To calculate unexpected uncertainty, this is multiplied with the predicted volatility $\Omega$ of the transition probability, governed by the free parameter $\omega$. In our model, where there are no additional volatility parents (see Supplement), the predicted volatility simplifies to:

$$\Omega = \exp(\omega) \tag{3}$$

This formulation follows directly from the belief update equations reported in the Supplement, where $\Omega$ determines the denominator of the predicted precision. Accordingly, unexpected uncertainty is calculated as:

$$U_{unexpected} = \mathrm{S}(\hat{\mu}_{x_{i,j}})\Big(1 - \mathrm{S}(\hat{\mu}_{x_{i,j}})\Big) \cdot \frac{1}{\pi_{x_{i,j}}} \cdot \exp(\omega) \tag{4}$$

In the probabilistic SRT task, response speed, rather than choice, serves as the primary measure reflecting participants' implicit learning of stimulus probabilities[3,75,76]. Accordingly, we modelled the log-transformed RTs using a linear regression that incorporated surprise ($\Im$), two types of uncertainty, and task-related binary predictors, namely Post-error (whether the trial followed an error-trial), and Post-reversal (whether the reversal had occurred). The linear regression (including an intercept and standard error of the regression σ) equation was as follows:

$$\mu\mathrm{RT} = \beta_0 + \beta_1.\Im + \beta_2.U_{expected} + \beta_3.U_{unexpected} + \beta_4.Post-error$$
$$+ \beta_5.Post-reversal \tag{5}$$

Finally, as with normal linear regression, the actual reaction times were sampled from a Gaussian $N$ with standard deviation $\sigma$:

$$\log \mathrm{RT} \sim N(\mu\mathrm{RT}, \sigma) \tag{6}$$

The categorical state-transition HGF with the linear regression-based response model for reaction times was built and fit using the Julia libraries HierarchicalGaussianFiltering.jl v0.5.4 (https://github.com/ComputationalPsychiatry/HierarchicalGaussianFiltering.jl) and ActionModels.jl v0.5.4 (https://github.com/ilabcode/ActionModels.jl), part of the TAPAS ecosystem for computational psychiatry[92]. Models were fit using Markov Chain Monte Carlo (MCMC) inference, specifically the NUTS() sampler[93] as implemented in Julia's probabilistic modelling ecosystem Turing.jl[94]. A model was fitted to the behaviour of each participant separately, using four chains with 2,000 samples each. Model convergence was confirmed through manual inspection (Supplement, Fig S1) and Gelman-Rubin $\hat{r}$ statistic <1.01[95,96].

Priors were chosen based on previous research that fit the HGF to a similar probabilistic four-choice SRT task[52] and have been reported in the supplement (Supplement, Table S2). Parameter recovery was tested on a simulated dataset and has been reported in the Supplement. To assess which model best fit the data, we fit the complete dataset to three different learning models. The model comparison procedure yielded the categorical state-transition HGF as the winning model (see Supplement for details of model selection and comparison).

### Reporting summary

Further information on research design is available in the Nature Portfolio Reporting Summary linked to this article.

## Data availability

The datasets analysed and modelled during the current study are publicly available: figshare.com/articles/dataset/Computational_signatures_of_uncertainty_are_reflected_in_motor_cortex_excitatory_neurochemistry/28430543/1[97]. Source data are provided with this paper.

## Code availability

The code used for the cognitive modelling, model-free analyses, and post hoc analyses are publicly accessible and can be found in the following repositories: github.com/naziajassim/hgf_srt[98] and github.com/naziajassim/computational_signatures_uncertainty[99].

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

## Acknowledgements

This work was funded by the Cambridge NIHR-Biomedical Research Centre [NIHR203312]. NJ was supported by the Newnham College April Trust PhD studentship and the Parke-Davis Postdoctoral Fellowship. The 7 T MRI was funded by an MRC Clinical Research Infrastructure Award [MR/M008983/1]. SBC received funding from the Wellcome Trust [214322\Z\18\Z], the Autism Center of Excellence at Cambridge, SFARI, the Templeton World Charitable Fund, the MRC and the NIHR Cambridge Biomedical Research Center. RPL received funding from the Wellcome Trust [G117272], and was supported by a Wellcome Trust Royal Society Henry Dale Fellowship [206691/Z/17/Z], an Autistica Future Leaders Award [ID: 7265], and a Lister Institute Prize Fellowship. The funders had no role in the design of the study; in the collection, analyses, or interpretation of data; in the writing of the manuscript, or in the decision to publish the results. For the purpose of open access, the author has applied a Creative Commons Attribution (CC BY) license to any Author Accepted Manuscript version arising from this submission. The neuroimaging analyses and the computational models were run using resources provided by the Cambridge Service for Data Driven Discovery (CSD3) operated by the University of Cambridge Research Computing Service (www.csd3.cam.ac.uk), provided by Dell EMC and Intel using Tier-2 funding from the Engineering and Physical Sciences Research Council (capital grant EP/T022159/1), and DiRAC funding from the Science and Technology Facilities Council (www.dirac.ac.uk).

## Author contributions

N.J. designed the study, acquired the funding with SBC and JS, collected the data, analysed the behavioural and neuroimaging data, visualised the task and results, ran the computational models, conducted the model comparisons, and drafted the manuscript. N.J. ran the HGF models with custom Julia scripts created by P.T.W. P.T.W. and C.M. designed the HGF model reported in the manuscript. P.T.W. created the HGF modelling scripts, conducted the parameter recovery simulations, and contributed to the modelling sections of the manuscript. OP programmed and piloted the task and contributed ideas for behavioural analyses. FHP contributed ideas for computational models, behavioural analyses, and visualisation of behavioural results, and contributed to the interpretation of results. C.R. and C.T.R. designed and tested the MRI protocol. S.B.C. and J.S. acquired the funding and contributed to the conception of the work. C.M. developed the HGF model and contributed to the interpretation of results. R.P.L. supervised the overall project. All authors reviewed the manuscript.

## Competing interests

The authors declare no competing interests.
