## [Transparent Peer Review file · Nature Communications]

Computational signatures of uncertainty are reflected in motor cortex excitatory neurochemistry

Corresponding Author: Dr Nazia Jassim

Version 0:

Reviewer comments:

Reviewer #1

(Remarks to the Author)

Key Findings:

Using a version of a probabilistic serial reaction time task, the authors first validated that participants had faster responses to high-probability trials than low-probability trials. They additionally found that, when splitting trials into three stages (early, middle, late), responses were faster initially in the post reversal period.

They developed a new implementation of the generalised HGF, to model probabilistic learning and categorical state-transitions in this task, and coupled it with a response model to explain reaction times with different predictors including HGF trajectories of key variables (e.g. surprise, different forms of uncertainty) using multiple regression. In the winning model (they tested a few learning + response models), the authors found that the dependent variable (reaction time estimate) was modulated significantly only by post-error slowing.

They suggest that non-significant effects from the other predictors may be due to large variation across participants in the other variables, exhibiting inconsistent effects on RT.

Regarding the neurochemical analysis, they found that higher M1 Glx levels correlated with larger prediction errors and lower volatility beliefs, suggesting its role in uncertainty processing. These findings exhibited some specificity as M1 GABA or Glx in the occipital cortex were not correlated with the computational variables.

They also examined whether trait anxiety modulated the general pattern of behavioural results. While high-trait-anxiety individuals were slower in general, they responded faster post-reversal. However, there was no association between volatility beliefs and trait anxiety levels, contrary to their predictions based on previous work.

Originality: The study introduces a new categorical state-transition HGF for modelling probabilistic learning in a four-choice task, extending beyond binary paradigms. They use the new framework of the generalised HGF, and the associated Julia implementation, which could be useful for further modelling studies in the field.

Assessing neurochemical markers via 7T MRS of learning in the task is novel and can inform future studies into the neurochemical basis of probabilistic learning under uncertainty, which could be expanded to the field of computational psychiatry by e.g. assessing similar processes in specific psychiatric conditions.

Overall, the methodological approach is original and robust, and the combination with neurochemistry analysis important. The analysis in trait anxiety seems not to be the focus of the current research, and given that the results are not very in line with hypotheses in anxiety research, potentially are less relevant to inform future work.

The work generally supports the main conclusions, but certain claims, such as the existence of individual “uncertainty fingerprints”, require stronger empirical support or more cautious framing. Some aspects of the data analysis, particularly the treatment of learning across binned “learning stages” instead of across trials would benefit from revision. It would also be beneficial to explain a bit better in the main text the key HGF free model parameter ω (it only became clear by looking at the Supplementary materials)

See suggestions for improvement in detail point-by-point:

Abstract

Clear and well-written, potentially the last sentence could be changed to be more targeted and specific, in line with the style of Nature Communications.

Introduction:

- line 35: The original HGF was not strictly a model of predictive coding. It was inspired by predictive coding but focused on volatility-coupling instead of value-coupling?

The recent extension of the HGF by Weber and colleagues aimed to precisely address this difference, formulating a generalised HGF able to assess predictive coding, including flexible value-coupling and volatility-coupling.

The authors refer to this extension later in the Introduction (lines 60-65). It would be helpful to make that distinction or clarify early on whether the authors directly refer to the generalised HGF in line 52. But the Weber et al 2024 reference is number 36, which is introduced later.

This distinction becomes clearer in the Discussion, but at this stage in the Intro it may lead to confusion in the readers. (Weber et al 2024 already open with this in their Abstract: " This includes predictive coding and hierarchical Gaussian filtering (HGF), which differ in the nature of hierarchical representations. Predictive coding assumes that higher levels in a given hierarchy influence the state (value) of lower levels. In HGF, however, higher levels determine the rate of change at lower levels. ...")

- The logical flow of the paragraph spanning lines 65-77 is not very clear, perhaps the authors can rephrase it to make the text more readable? The text in this paragraph moves from sensorimotor learning to glutamate to reversal learning to uncertainty, but the logical connection is missing.

- It would be important to clarify in the introduction whether learning in this study refers to probabilistic learning, and motor responses are just a readout of behaviour reflecting learning (lines 78-86), or whether it does indeed study sensorimotor learning. The Introduction mentions sensorimotor learning a couple of times, but it is not clear that this is what will be studied (despite explicitly mentioning "sensorimotor probabilistic reversal learning"). Figure 1 explains this better, but it would be helpful to make this more explicit briefly in the Introduction (e.g. mentioning that there are two sequences of button presses that have low or high probability of occurrence). Only after reading the caption in Figure 1 and reaching the end of the Introduction, and start of Results, was it clear which behaviour the authors are studying.

- Figure 2. Please check whether for Nature journals it is recommended (or required) that sample size of the sample used for the displayed results is indicated. Potentially also statistical results could be briefly mentioned, along with the type of test.

Results

- page 7, LME, linear mixed models could be very useful at assessing learning across trials. Did the authors implement LMEs with trial as predictor? Did they split the trials into three phases per session a priori, based on hypotheses or otherwise what was the rationale for this split?

What would an LME with trial (instead of stage: early, middle, late) reveal? Is there a robust effect of probability on learning across trials?

- HGF Results. Potentially referring to "belief distributions" is more suitable than "beliefs", at least when describing the summary statistics of Gaussian belief distributions via mean and precision variables.

- Free parameter ω : "The learning of transition probabilities is contingent on the parameter ω , which represents the expectation of general volatility in the environment. " For HGF users this will be a moment in the manuscript where they would need to see an expression defining ω (or a reference to Methods). This is missing from the main manuscript and Methods. See my comment below.

Given that this parameter features prominently in the analysis and results, having it defined or at least included in an equation would be beneficial.

This is explained after equation S5 in Supp Materials:

"observational noise, here it is infinite (i.e., the denominator only consists of the predicted volatility Ω) . As there are no additional volatility parents, Ω only consists of the ω parameter - the general expected volatility of the environment. Note that, as ω is shared between all ..."

Discussion:

- "uncertainty fingerprints": While this aspect is clearly interesting in contemporary research, the current manuscript does not robustly demonstrate that individuals indeed have unique uncertainty fingerprints. We can see variation in different parameters across participants, as in any study, however, I am not convinced that inter-individual variation is equivalent to evidence for individual fingerprints. If the authors consider that individuals have robust, stable individual uncertainty fingerprints, they should provide this evidence, which currently is lacking. Alternately, they can argue that the observed variation may be interesting to explore further in future work aiming to identify uncertainty fingerprints. Accordingly, without further evidence, I think the emphasis on "fingerprints" should be relaxed in the Intro and Discussion.

Lines 289-291 seem more balanced in interpreting the results: "By linking computational estimates of uncertainty processing with neurochemical data, we provide a deeper understanding inter-

individual differences in adaptive behaviour in unpredictable environments. “

Methods:

- LME models page 19: As suggested above, one of the strengths of linear mixed models is the robust assessment of changes in a DV across trials, and including participants as random effects on slopes, in addition to random effects on the intercept would be a key candidate model that should be considered. Such a model would build on LME Model 2 (or could replace it).

Thus, two models with/without random effects of participants on slopes could be considered.

- Similarly, a LME model of surprise (as shown in Supp Materials) could be run using trials as predictor, instead of learning stage, and with/without random effects on slope and intercept. Such models could reveal intercept estimates for surprise in the pre and post-reversal, as well as identify whether a negative slope for surprise vs trials is found in the pre-reversal period, but not in the post-reversal (e.g. slope estimate with a CI including zero). This would be interesting to establish whether surprise decreased with learning in the pre-reversal phase but not in the post-reversal phase. Why that is the case and what that means for learning categorical stage-transitions in this study could also be discussed.

- Lines 442-445: Generalised HGF equations: The free model parameters ω is not defined or included in Equation 3. To avoid confusion, the authors could indicate that this parameter appears in Supp Materials. On the other hand, given that this parameter is important for understanding the results, it would be important to briefly define it or introduce it in the equations of the main manuscript.

-line 465: The current recommendation for Gelman-Rubin $\hat{\sigma}$ statistic is 1.01, not 1.1. Which values did the authors obtain? Vehtari A, Gelman A, Simpson D, Carpenter B, Bürkner PC. Rank-normalization, folding, and localization: An improved $R^{\hat{\sigma}}$ for assessing convergence of MCMC (with discussion). Bayesian analysis. 2021 Jun;16(2):667-718.

Minor:

Some small typos include

- adding stop sign before closing "" statements. → line 31 refer to as “uncertainty fingerprints.”
- missing en-dashes (some en-dashes are followed by a dash at the closing clarification): lines 47-48: “Minimising these prediction errors – as is central to predictive coding models 31 - may be”
- l175: β_4 . Post – error is provided as (0.10, 0.07), but this may be the reverse? (0.07, 0.10)?
- l310: pparticipants
- Supp Materials: p6: “agonistic” should be perhaps agnostic?

(Remarks on code availability)

The Rmd files contain enough annotations for readers to understand the analyses steps. These could be useful for future researchers.

I only checked the Matlab files briefly, but they seem in line with my knowledge of Psychtoolbox. I did not run the code, though.

README has only minimal information and directs the reader to the paper. It would be useful to mention which aspects of the analysis are included in the Rmd files. While this is obvious once one checks the Rmd files, it could be useful to have an outline at this top hierarchical level represented by the README file.

Thanks

Reviewer #2

(Remarks to the Author)

This is a really interesting paper testing out an exciting concept: linking computational measures of uncertainty to neurochemical data. I was very pleased to review it! The main results are the use of a novel version of the HGF, seeing which elements slow RTs on the SRT task (in this case, just reversal), and linking the results from this to Glx levels in M1, measured using 7T MRS. It is also useful to see the lack of replication of the trait-anxiety/volatility finding, which will add to computational psychiatry literature. However, I had a few concerns. Firstly, I was not sure to what extent the specific tests were planned. For instance, I don't know what 'high-level delta' is, at which timepoint it is calculated, or why that was chosen as a variable to relate to Glx level. The argument of the rest of the paper is that you get an 'uncertainty fingerprint', but 'high-level delta' isn't part of that.

I also felt that the 'uncertainty fingerprints' that are derived are not really used or discussed - I think these are a really important contribution. Similarly, other parts of the introduction emphasise certain ideas (e.g. non-binary choices), but this narrative is not continued through the paper.

The introduction in general seems quite technical - more work could be done to further justify the research question and summarise extant literature.

My understanding was that at 7T, glutamate and glutamine can be resolved separately (e.g. Lally et al. 2016). Was this not

possible using this sequence?

Minor details

The description of the HGF is generally very clear, but it would be good to know how many higher-order nodes there are and how this is determined (the figure just shows 'a multiplicity' of them).

Why is subject ID displayed in figure 4's correlelogram? This is a little confusing to me.

There are some typos in the methods section - e.g. 'pparticipants' and saying that sequence B is both the most and least probable in session 2, 'assessed BY means...' on line 391.

I was not able to access the dataset through the figshare link.

Figure S1 is a little unclear - what are the error bars in plot A? Equally, Figure S2 is fairly grainy - and in the description, you say that surprise increases in the middle and late parts of the post-reversal session, but the graph (which is binned differently) doesn't reflect this.

(Remarks on code availability)

The code accessible in the github seems to use pre-cleaned data, it would be good if the data cleaning code was available too. Equally, the code to launch the models wasn't included.

The data on figshare was not available, so I was not able to see whether the code would be a usable resource.

Reviewer #3

(Remarks to the Author)

I enjoyed reading this paper by Jassim and colleagues, in which they report associations between probabilistic reinforcement learning processes and glutamate/glutamine assessed using magnetic resonance spectroscopy. They demonstrate that prediction errors and beliefs about environmental volatility during a motor learning task are linked to Glx levels (a compound measure of glutamate and glutamine) in the motor cortex, suggesting that glutamatergic neurotransmission in this region may be involved in motor learning.

The study is innovative and the methods appear robust, while the paper is well-written (save for some points of clarification that may assist the reader). Below I note some suggestions that could be incorporated into a revision.

1. The computational modelling is somewhat difficult to understand as it seems to diverge somewhat from typical applications of computational modelling to probabilistic learning tasks. First, it is unclear why the model was fit to log RT rather than to responses directly, which would presumably provide a clearer readout of how participants are learning about the probability of each item in the sequence. Second, the regression approach to model fitting seems like a somewhat indirect way to infer how participants are estimating uncertainty. Ordinarily we might expect the model itself to have free parameters (e.g., learning rates) that can capture different learning profiles depending on their value; here it seems that the learning model itself is fairly rigid (aside from the expectation of volatility) and instead it is the extent to which this rigid model predicts participants' behavior that is being estimated. Does this affect the interpretation of the results?
2. For the comparison models reported in the supplementary material, how were these fit to the data? Did this use a similar regression-style approach?
3. The concept of volatility could be introduced a little better within the context of the present task. Since it does not feature a clear volatility manipulation (aside from the reversal), it can be a little hard to understand why we might be expecting participants to track volatility at all.
4. Is there a reason for using average prediction errors in the MRS analyses rather than surprise, given that surprise seems more relevant to the Bayesian updating process?
5. Could the PE-Glx effect be confounded by performance? I.e., worse performers would see greater prediction errors, and performance itself may be related to Glx.
6. Given MRS was performed at 7T, it should be feasible to separate glutamate from glutamine rather than using the combined Glx measure, was this attempted?
7. How far apart in time were the scan and the behavioral testing?
8. Some of the phrasing could be toned down a little to avoid overstating the results. For example p16 "M1 Glx plays a critical role" – it is hard to say from a correlational study like this whether it really is critical.
9. A technical question: for a task that involves estimating probabilities, it seems like using multiple Gaussian distributions to represent these probabilities is a little awkward. Is there any way the model could be respecified to use a Dirichlet distribution that would more easily capture these values? I don't necessarily expect the authors to do this, it's more a point of curiosity.
10. Minor point: p9, it may be worth rephrasing "these beliefs are updated faster" as it can be easy to misread "faster" as meaning "faster in time" within the context of reaction times and post-error slowing.

(Remarks on code availability)

Reviewer #4

(Remarks to the Author)

The paper titled "Neurochemical markers of uncertainty processing in humans" by Jassim and colleagues presents a unique investigation linking computational modelling and MR spectroscopy. Specifically, the authors use a novel variant of the Hierarchical Gaussian Filter to estimate volatility in a serial reaction time task and link it to behaviour as well as glutamate/glutamine levels in the motor cortex. Additionally, they associate individual differences in trait anxiety to acceleration of reaction times after reversal.

This investigation constitutes a well-executed and technically sophisticated study generating novel insights into the neurochemical mechanisms of uncertainty. Below I present a list of questions and suggestions - I would be happy to recommend the paper for publication if these can be addressed. Though, ultimately, I leave it up to the authors which they choose to incorporate in their paper.

Title - I think the title should more closely reflect the anatomical specificity of the analyses, i.e. it should mention i) motor cortex and ii) glutamate (Glx). As it stands, it gives the impression that a broad range of neurochemical systems and regions has been investigated.

It is unclear why M1 was selected as the target region for MRS. Other regions, such as the OFC, ACC, striatum etc (e.g. Behrens et al., 2007; Muller et al., 2019) have been more consistently linked to uncertainty, and they would be more likely a priori candidates.

When discussing the Glx concentrations in relation to high-level PE and volatility beliefs, could the authors add whether such PEs and volatility estimates correlate? This is to clarify whether the two are related or whether the glutamate levels are related to both independently. If they relate, this could also be added to the discussion.

In the introduction, I think it might help readers to have a bit more introduction into the topic of uncertainty. Specifically, what is volatility and unexpected/expected uncertainty and how they are related, following from a nice overview in Sandhu, Xiao & Lawson 2023.

On a related point, I would suggest toning down the use of "uncertainty fingerprints" as it gives the impression that an extensive array of uncertainty measures has been used. While several aspects of learning under uncertainty are estimated, this is by no means exhaustive.

Task - it is somewhat unclear whether unlikely transitions (0.15) lead to a position within the same sequence or whether they lead to sequence-to-sequence transitions. I was assuming that the transitions are within-sequence, especially given the high accuracy.

Figure 3c - this point may stem from my lack of understanding of aspects of the model - why are the state transition probabilities roughly equal for the three non-current options [~ 0.3 , ~ 0.3 , ~ 0.3 , 0]? Shouldn't learning lead them to reflect the true transition probabilities [0.85, 0.15, 0, 0]?

Reaction times are often believed to reflect aspects of uncertainty processing. Here, however, RTs do not seem to reflect model-based estimates. While this is made clear in figure 4a, it might also deserve a minor discussion point.

In the introduction, the selection of TA is not directly justified using past literature. It is also not clear why more specific constructs were not included (Intolerance of Uncertainty Scale).

At a number of points, the authors refer to a specific signal as a "reversal" e.g. M1 Glx on page 12 (line 187). However, reversal here cannot be dissociated from simple temporal effects. I.e., if there is an interaction between session and Glx, this could be that Glx levels are more closely associated with RTs in one of the sessions. Please correct this to prevent the wrong impression that the signal is specific contingency reversals.

The "high-level PE" reflects general accuracy of predictions across all positions. I was just wondering if it could be described (perhaps in discussion) in terms of information gain across the entire belief state, in order to provide an intuitive understanding of this aggregate measure. On a related note, were the averaged prediction errors signed or unsigned? In my understanding they should be unsigned otherwise positive and negative PEs cancel each other and result in no "high level PE".

Regarding TA and uncertainty - while some past work has linked TA to uncertainty processing (Yan et al., 2025; Browning et al. 2015), a series of recent papers employing large sample sizes have not replicated these results (Satti et al., 2025; Suddell et al., 2024). The authors show a lack of relationship between TA and volatility, expected and unexpected uncertainty. I think adding a correlation matrix of these measures to supplementary would be helpful for the ongoing debate on the topic.

https://pure.mpg.de/rest/items/item_3592172/component/file_3634933/content
<https://doi.org/10.31234/osf.io/hm46n>

Please state in Results which questionnaire was used to assess TA - I know it's in the Methods but it would help with readability.

Along similar lines, when reading the Results, I was missing basic information about i) the sample used and its size; ii) the general design of the study (this might give the impression that MRS was acquired simultaneously with the task).

Exclusions - individual trials with RTs longer than 2 SDs were excluded (approx 4.5% of trials). What was the justification for this choice?

Some references are in APA format, i.e. not numbered: Mathys 2011,2014; Weber 2024; Marshall 2016.

At least one point (p 14, line 227), the authors refer to "changes in volatility". This is not accurate as there was only a single reversal.

(Remarks on code availability)

Version 1:

Reviewer comments:

Reviewer #1

(Remarks to the Author)

The authors have done an excellent job addressing the queries, the text now reads better and the new analyses (e.g. LME) strengthen the results, having a more principled way of assessing $\log(\text{RT})$ and surprise across trials than by using the (potentially arbitrary) three-way split of time into segments. I recommend the manuscript for publication.

(Remarks on code availability)

README file has improved.

Reviewer #2

(Remarks to the Author)

The revisions made are extensive and have improved what was already an excellent paper. The authors should just double-check they've not introduced typos (e.g. should read 'evolution OF surprise' in response to R1.11), and 'three OF the comparison models' in response to R3.03.

The rationale for the use of high-level delta was appreciated, thank you. It may be useful to relate surprise to Glx , too, as an exploratory analysis, and to aid our understanding of the specificity of the PE findings.

My question was also to what extent this study was preregistered - am I correct to assume it was not?

I am also satisfied with the explanation of the use of Glx not Glu/Gln separately. I think the inclusion of this explanation in the Discussion is also useful, given that I was not the only reviewer with this query! It may be worth briefly mentioning what indices you use of 'quantification uncertainty' in the text, too.

Why are the colours so much darker in the novel correlation analysis without subjID? This implies the magnitudes of correlations have increased?

Thank you for clarifying the script locations. It would also be useful to link the two githubs together in some way - even by linking from one to the other in the 'readme' files.

(Remarks on code availability)

Reviewer #3

(Remarks to the Author)

The authors have addressed my comments thoroughly. This is a very interesting paper that I am sure will be of interest to many, and I look forward to seeing it published.

(Remarks on code availability)

Reviewer #4

(Remarks to the Author)

Thank you for addressing the comments. I have no further suggestions or concerns.

(Remarks on code availability)

I reviewed the code but I didn't run it. It looks sensible.

Computational signatures of uncertainty are reflected in motor cortex excitatory neurochemistry

Corresponding author: Nazia Jassim

Response to reviewers

We sincerely thank all four reviewers for their time and thoughtful evaluation of our manuscript. We were encouraged by the uniformly positive and constructive feedback, and appreciate the reviewers' recognition of the significance and quality of our work. We believe the revisions we have made in response to their comments have further strengthened the manuscript. Below, we provide a point-by-point response to each comment. Original reviewer comments are shown in **blue**, our responses in **bold black** font, and revised text is indicated in *italics*, with page and line numbers provided for reference. All changes in the revised manuscript are highlighted in **yellow**.

REVIEWER COMMENTS

Reviewer #1 (Remarks to the Author):

Key Findings:

Using a version of a probabilistic serial reaction time task, the authors first validated that participants had faster responses to high-probability trials than low-probability trials. They additionally found that, when splitting trials into three stages (early, middle, late), responses were faster initially in the post reversal period. They developed a new implementation of the generalised HGF, to model probabilistic learning and categorical state-transitions in this task, and coupled it with a response model to explain reaction times with different predictors including HGF trajectories of key variables (e.g. surprise, different forms of uncertainty) using multiple regression. In the winning model (they tested a few learning + response models), the authors found that the dependent variable (reaction time estimate) was modulated significantly only by post-error slowing. They suggest that non-significant effects from the other predictors may be due to large variation across participants in the other variables, exhibiting inconsistent effects on RT. Regarding the neurochemical analysis, they found that higher M1 Glx levels correlated with larger prediction errors and lower volatility beliefs, suggesting its role in uncertainty processing. These findings exhibited some specificity as M1 GABA or Glx in the occipital cortex were not correlated with the computational variables. They also examined whether trait anxiety modulated the general pattern of behavioural results. While high-trait-anxiety individuals were slower in general, they responded faster post-reversal. However, there was no association between volatility beliefs and trait anxiety levels, contrary to their predictions based on previous work.

Originality: The study introduces a new categorical state-transition HGF for modelling probabilistic learning in a four-choice task, extending beyond binary paradigms. They use the new framework of the generalised HGF, and the associated Julia implementation, which could be useful for further modelling studies in the field. Assessing neurochemical markers via 7T MRS of learning in the task is novel and can inform future studies into the neurochemical basis

of probabilistic learning under uncertainty, which could be expanded to the field of computational psychiatry by e.g. assessing similar processes in specific psychiatric conditions. Overall, the methodological approach is original and robust, and the combination with neurochemistry analysis important. The analysis in trait anxiety seems not to be the focus of the current research, and given that the results are not very in line with hypotheses in anxiety research, potentially are less relevant to inform future work.

The work generally supports the main conclusions, but certain claims, such as the existence of individual “uncertainty fingerprints”, require stronger empirical support or more cautious framing. Some aspects of the data analysis, particularly the treatment of learning across binned “learning stages” instead of across trials would benefit from revision. It would also be beneficial to explain a bit better in the main text the key HGF free model parameter ω (it only became clear by looking at the Supplementary materials).

We are grateful for this detailed summary of our work and for the reviewer’s positive view of its originality and contribution to the field. We address each of their comments in detail below.

See suggestions for improvement in detail point-by-point:

[R1.01]

Abstract

Clear and well-written, potentially the last sentence could be changed to be more targeted and specific, in line with the style of Nature Communications.

Thank you very much for the positive feedback. In line with your suggestion, we have changed the last sentence of the abstract to be more specific:

“This study establishes a direct neurochemical correlate of hierarchical belief updating, identifying motor cortex glutamate + glutamine as an important neural marker of inter-individual differences in uncertainty processing.”

Introduction

[R1.02]

line 35: The original HGF was not strictly a model of predictive coding. It was inspired by predictive coding but focused on volatility-coupling instead of value-coupling? The recent extension of the HGF by Weber and colleagues aimed to precisely address this difference, formulating a generalised HGF able to assess predictive coding, including flexible value-coupling and volatility-coupling. The authors refer to this extension later in the Introduction (lines 60-65). It would be helpful to make that distinction or clarify early on whether the authors directly refer to the generalised HGF in line 52. But the Weber et al 2024 reference is number 36, which is introduced later.

This distinction becomes clearer in the Discussion, but at this stage in the Intro it may lead to confusion in the readers.(Weber et al 2024 already open with this in their Abstract: “ This includes predictive coding and hierarchical Gaussian filtering (HGF), which differ in the nature of hierarchical representations. Predictive coding assumes that higher levels in a given hierarchy influence the state (value) of lower levels. In HGF, however, higher levels determine the rate of change at lower levels. ...”)

Thank you for this opportunity to clarify the distinction between different implementations of the HGF. We agree that the original HGF, as introduced by Mathys et al. (2011), was not strictly a model of predictive coding. As you point out, it was inspired by hierarchical Bayesian inference and predictive coding principles; however its core mechanism relied on volatility coupling, where higher levels in the hierarchy modulate the rate of change (i.e., learning rate) at lower levels, rather than directly influencing their state values. In contrast, predictive coding frameworks typically assume value coupling, whereby higher-level states directly shape lower-level state estimates. The recent generalised HGF proposed by Weber et al. (2024) introduces a flexible framework capable of modelling both volatility and value coupling.

We have now made this distinction clearer in the revised introduction (Page 4 , lines 60-71):

“The Hierarchical Gaussian Filter (HGF) is a Bayesian model of adaptive learning that estimates hidden, dynamically changing states of the environment based on noisy observations ^{6,35}. These estimates correspond to probabilistic “beliefs” across multiple hierarchical levels: the lowest level represents raw sensory input, while higher levels capture more abstract beliefs about the structure and volatility of the environment. In its original formulation, the HGF employed volatility coupling, wherein higher levels modulate the rate of change (i.e., learning rate) at lower levels ⁶. However, classic predictive coding assumes that higher levels predict the value of lower levels (i.e, value coupling). To formalise this distinction, the recent generalised version of the HGF introduces a network-based formulation that allows for both volatility coupling and value coupling between hierarchical belief states ³⁶. In this extended framework, each belief is represented as a probabilistic node, and higher-level nodes can influence both the drift (value) and volatility (rate of change) of lower-level nodes, enabling a richer and more flexible modelling of adaptive learning ³⁶.”

[R1.03]

The logical flow of the paragraph spanning lines 65-77 is not very clear, perhaps the authors can rephrase it to make the text more readable? The text in this paragraph moves from sensorimotor learning to glutamate to reversal learning to uncertainty, but the logical connection is missing.

Thank you for this suggestion to improve readability. We have revised this paragraph based on this comment and your subsequent comment. Please see our response to your next comment for details of the updated text.

[R1.04]

It would be important to clarify in the introduction whether learning in this study refers to probabilistic learning, and motor responses are just a readout of behaviour reflecting learning (lines 78-86), or whether it does indeed study sensorimotor learning. The Introduction mentions sensorimotor learning a couple of times, but it is not clear that this is what will be studied

(despite explicitly mentioning “sensorimotor probabilistic reversal learning”). Figure 1 explains this better, but it would be helpful to make this more explicit briefly in the Introduction (e.g. mentioning that there are two sequences of button presses that have low or high probability of occurrence). Only after reading the caption in Figure 1 and reaching the end of the Introduction, and start of Results, was it clear which behaviour the authors are studying.

Thank you for this suggestion. We agree that the discussion of sensorimotor and probabilistic learning could be more seamlessly integrated to clarify the specific learning behaviour under investigation. To address this, we have revised the paragraph to clarify that laboratory-based tasks probe implicit probabilistic learning through sensorimotor responses. We also introduce our task briefly here, rather than later in the introduction, and explain that motor responses serve as behavioural readouts of implicit learning of the probabilistic sequences. This should make it clearer from the outset that we are studying probabilistic learning, with motor responses as a proxy for learning performance.

The revised paragraph reads as follows (Page 4-5, lines 72-88):

“The brain uses probabilistic models to optimise performance and guide responses during learning^{37–39}. Laboratory-based tasks of implicit probabilistic learning are designed to probe how participants acquire the underlying probabilistic structure without explicit awareness. In the present study, participants perform a sensorimotor task involving two sequences that differ in their probability of occurrence, with one occurring frequently and the other less so (Fig 1). The task examines how individuals implicitly learn the underlying probabilistic structure, using motor output as a proxy for learning performance. As sensory evidence is accumulated, it is prominently represented in motor cortical areas to guide task-relevant actions^{40,41}. At the cellular level, as learning progresses, primary motor cortex neurons (M1) neurons undergo adaption and glutamate-driven plasticity^{42,43}. Specifically, the primary excitatory neurotransmitter glutamate plays a critical role in rapid inter-cellular communication to facilitate learning and choice behaviour⁴⁴. Research in rodents has shown that learning in dynamic environments—as measured through reversal learning paradigms—is strongly dependent on glutamatergic modulation^{45–48}. With the advent of Magnetic Resonance Spectroscopy (MRS), a non-invasive technique to measure tissue metabolites in vivo, the metabolites related to probabilistic learning can now be studied effectively in living humans^{49,50}.”

[R1.05]

Figure 2. Please check whether for Nature journals it is recommended (or required) that sample size of the sample used for the displayed results is indicated. Potentially also statistical results could be briefly mentioned, along with the type of test.

Thank you for bringing this to our attention. According to the journal’s guidelines, “Figure legends should be <350 words each. They should begin with a brief title sentence for the whole figure and continue with a short statement of what is depicted in the figure, not the results (or data) of the experiment or the methods used. Legends should be detailed enough so that each figure and caption can, as far as possible, be understood in isolation from the main text.”

Although not explicitly required by the journal, we agree that including the sample size can aid interpretation of the plots and have now added this information to the caption. We have

not included details of statistical tests, as the figure displays summary statistics rather than results from the LME models reported in the main text.

[R1.06]

page 7, LME, linear mixed models could be very useful at assessing learning across trials. Did the authors implement LMEs with trial as predictor? Did they split the trials into three phases per session a priori, based on hypotheses or otherwise what was the rationale for this split? What would an LME with trial (instead of stage: early, middle, late) reveal? Is there a robust effect of probability on learning across trials?

Thank you for this suggestion. We used learning stages (early, middle, late) rather than trial number in our primary analyses to ease interpretability of the results. This division was decided a priori based on prior studies using similar tasks, which often report learning dynamics across defined stages rather than trial-level effects (e.g., Frangou et al., 2019, Collins et al., 2012)

However, we agree that trial number may provide a more sensitive and continuous measure of learning dynamics than the categorical learning stage-based approach. Thus, as per your recommendation, we have now replaced the LME model using Trial Number as a continuous predictor (centred within session), along with Stimulus probability (High vs Low probability) and Session (Pre vs Post-reversal), as well as their interactions. This allowed us to examine learning as a continuous process across trials. Based on your other recommendation (under Methods) to include random slopes for participants in in this LME model, we have now revised the model as follows:

$$\log(RT) \sim (\text{Stimulus probability} + \text{Session}) * \text{Trial Number} \\ + (1 + \text{Trial Number} | \text{Participant})$$

These results now included in the manuscript (Results, Page 7, Lines 132-142):

“To examine learning on a finer timescale, we also modelled trial-by-trial changes in log-transformed RT (LME Model 2). This model included fixed effects for stimulus probability (High vs Low Probability), session (Pre- vs Post-reversal), trial number, as well as all interactions. Crucially, it also included random intercepts and random slopes for trial number by participant to account for inter-individual differences in learning trajectories. We found a significant interaction between stimulus probability and trial number. Specifically, for low probability stimuli, responses slowed over time compared to high probability stimuli ($b = 0.0097$, $SE = 0.0019$, $t(79,160) = 5.22$, $p = 1.83 \times 10^{-7}$). The interaction between session and trial number was also significant ($b = 0.0172$, $SE = 0.0013$, $t(79,160) = 13.06$, $p < 2 \times 10^{-16}$). These findings suggest that participants differed meaningfully in their trial-by-trial learning and adjusted their responses based on both stimulus probability and task session.”

Please also see our response to your other comment [R1.10] for details about the model comparison of the trial-by-trial learning LME models with and without random slopes for participants.

- Frangou, P., Emir, U.E., Karlaftis, V.M. et al. Learning to optimize perceptual decisions through suppressive interactions in the human brain. Nat Commun 10, 474 (2019). <https://doi.org/10.1038/s41467-019-08313-y>
- Collins AG, Frank MJ. How much of reinforcement learning is working memory, not reinforcement learning? A behavioral, computational, and neurogenetic analysis. Eur J Neurosci. 2012 Apr;35(7):1024-35. doi: 10.1111/j.1460-9568.2011.07980.x.

[R1.07]

HGF Results. Potentially referring to “belief distributions” is more suitable than “beliefs”, atleast when describing the summary statistics of Gaussian belief distributions via mean and precision variables.

We agree with your suggestion. To be more precise, we have replaced “beliefs” with “belief distributions” where required.

For example, Page 9, lines 169-179 of the results:

“The HGF holds Gaussian belief distributions about $x_{i,j}$, with μ representing the mean of the current belief distribution about the transition probability and the precision π representing the certainty in that belief distribution (see Fig 3b-c for an example of how these two values evolve over time for a specific transition probability). Belief updates are larger in magnitude when prediction errors are large, and when the precision π of the current belief distribution is low. At each trial, the means μ of the belief distribution are transformed to probability space and normalised to construct the final expected transition probabilities (Fig 3d displays an example session where these 16 transition probabilities are learnt over time). The learning of transition probabilities is contingent on the parameter ω , which represents the expectation of general volatility in the environment (Methods, Equation 3). Higher values of ω lead to less certainty and faster updating of beliefs.”

Figure 3 caption:

“.. Evolution of higher-level belief distributions μ (top graph) about transition probabilities and the corresponding posterior precision π (bottom graph) at the continuous node ($x_{i,j}$) for a specific transition (namely, from Category 1 at trial $t-1$ to Category 4 at trial t), at an example session. Each data point corresponds to the mean of the state transition belief distribution (μ) or posterior precision (π) in bins of 120 trials (x -axis), across each session of the task (Pre-reversal in green, and Post-reversal in orange).”

[R1.08]

Free parameter ω : “The learning of transition probabilities is contingent on the parameter ω , which represents the expectation of general volatility in the environment. “ For HGF users this will be a moment in the manuscript where they would need to see an expression defining ω (or a reference to Methods). This is missing from the main manuscript and Methods. See my comment below.

Given that this parameter features prominently in the analysis and results, having it defined or at least included in an equation would be beneficial. This is explained after equation S5 in Supp Materials:

“observational noise, here it is infinite (i.e., the denominator only consists of the predicted volatility Ω) . As there are no additional volatility parents, Ω only consists of the ω parameter

- the general expected volatility of the environment. Note that, as ω is shared between all ...”

Thank you for this suggestion. We agree that providing an explicit definition of the free parameter ω in the main Methods section is important for clarity, particularly given its role in shaping predicted volatility and consequently, unexpected uncertainty. We have now amended the Methods section to clarify that in our model, predicted volatility $\Omega = \omega$, as there are no additional volatility parents. We also now include this in equation form (revised Equation 3). In the unexpected uncertainty calculation (revised Equation 4), we replace $\exp(\Omega)$ with $\exp(\omega)$ to make the dependency on the free parameter explicit in the main manuscript.

Methods, Page 24, lines 551-560:

“To calculate “unexpected” uncertainty, this is multiplied with the predicted volatility Ω of the transition probability, governed by the free parameter ω . In our model, where there are no additional volatility parents (see Supplement), the predicted volatility simplifies to:

$$\Omega = \exp(\omega)$$

Equation 1

This formulation follows directly from the belief update equations reported in the Supplement, where Ω determines the denominator of the predicted precision. Accordingly, unexpected uncertainty is calculated as:

$$U_{\text{unexpected}} = S(\hat{\mu}_{x_{i,j}}) \left(1 - S(\hat{\mu}_{x_{i,j}})\right) \cdot \frac{1}{\pi_{x_{i,j}}} \cdot \exp(\omega)$$

Equation 2”

Discussion:

[R1.09]

“uncertainty fingerprints”: While this aspect is clearly interesting in contemporary research, the current manuscript does not robustly demonstrate that individuals indeed have unique uncertainty fingerprints. We can see variation in different parameters across participants, as in any study, however, I am not convinced that inter-individual variation is equivalent to evidence for individual fingerprints. If the authors consider that individuals have robust, stable individual uncertainty fingerprints, they should provide this evidence, which currently is lacking. Alternately, they can argue that the observed variation may be interesting to explore further in future work aiming to identify uncertainty fingerprints. Accordingly, without further evidence, I think the emphasis on “fingerprints” should be relaxed in the Intro and Discussion. Lines 289-291 seem more balanced in interpreting the results: “By linking computational estimates of uncertainty processing with neurochemical data, we provide a deeper understanding inter-individual differences in adaptive behaviour in unpredictable environments. “

Thank you for this thoughtful observation, which is also echoed by Reviewer #4 [R4.05]. We acknowledge that while our findings show behavioural, computational, and neurochemical evidence for inter-individual variation in uncertainty-processing, we do not provide direct evidence for stable “uncertainty fingerprints” across time or contexts. To reflect this, we have revised the manuscript to soften our interpretation and avoid suggesting that such fingerprints are definitively established. Specifically, we have adjusted the terminology throughout, replacing “uncertainty fingerprints” with “individual differences in uncertainty processing”. Some examples of revised sentences:

Abstract:

Original: *“Here we use computational phenotyping to examine individual differences in “uncertainty fingerprints” in relation to neurometabolites and trait anxiety in humans.”*

Revised: *“Here we use computational phenotyping to examine inter-individual differences in uncertainty processing in relation to neurometabolites and trait anxiety in humans.”*

Introduction:

Original: *“Consequently, it is crucial to not only understand the neurocomputations of uncertainty processing, but also to use computational phenotyping to identify hidden variables that account for individual differences — what we refer to as ‘uncertainty fingerprints.’ ”*

Revised: *“Consequently, it is crucial to not only understand the neurocomputations of uncertainty processing... to identify hidden variables that account for individual differences.”*

Discussion:

Original: *“Notably, at the participant-level, ... revealing distinct “uncertainty fingerprints” ”*

Revised: *“Notably, at the participant-level, ... revealing meaningful evidence of individual variability in uncertainty-related computations.”*

[R1.10]

Methods:

LME models page 19: As suggested above, one of the strengths of linear mixed models is the robust assessment of changes in a DV across trials, and including participants as random effects on slopes, in addition to random effects on the intercept would be a key candidate model that should be considered. Such a model would build on LME Model 2 (or could replace it). Thus, two models with/without random effects of participants on slopes could be considered.

We appreciate this suggestion regarding the inclusion of random slopes in the trial-by-trial learning LME model. In line with this, we re-specified LME Model 2 to include random slopes for trial number by participant, allowing us to account for individual differences in learning trajectories across trials. Specifically, the revised model included both random intercepts and random slopes for Trial Number:

$\log(RT) \sim (\text{Stimulus probability} + \text{Session}) * \text{Trial Number} + (1 + \text{Trial Number} | \text{Participant})$

We compared this model to the original one, which included only random intercepts, using likelihood ratio tests, AIC, and BIC. The random-slope model showed a significantly better fit ($\chi^2(2) = 641.36, p < .001$) and also had lower AIC (-41511 vs -40874) and BIC (-41418 vs -40800) values. This suggests that individuals differ meaningfully in their trial-by-trial learning, justifying the inclusion of random slopes in the final model. Thus, based on your recommendation and the model comparison results, we have replaced the original LME Model 2 in the manuscript with the model including participants as random effects on slopes in addition to random effects on the intercept.

Further, in our revisions, we report details of the LME model comparisons for this analysis, as well as for the one addressed in your following comment. These have been reported in page 6 of the revised Supplement.

[R1.11]

Similarly, a LME model of surprise (as shown in Supp Materials) could be run using trials as predictor, instead of learning stage, and with/without random effects on slope and intercept. Such models could reveal intercept estimates for surprise in the pre and post-reversal, as well as identify whether a negative slope for surprise vs trials is found in the pre-reversal period, but not in the post-reversal (e.g. slope estimate with a CI including zero). This would be interesting to establish whether surprise decreased with learning in the pre-reversal phase but not in the post-reversal phase. Why that is the case and what that means for learning categorical stage-transitions in this study could also be discussed.

Thank you for this suggestion. Following your recommendation, we conducted an additional LME analysis modelling surprise (\mathfrak{S}) as a function of trial number, session, and their interaction, including random intercepts and slopes for trial number by participant:

$\mathfrak{S} \sim \text{Session} * \text{Trial Number} + (1 + \text{Trial Number} | \text{Participant})$

The results revealed a significant negative slope for surprise across trials during the pre-reversal session ($b = -0.01175, SE = 0.0024, p < 0.001$), indicating that surprise decreased as participants learned the task structure. In contrast, the slope for surprise during the post-reversal session (i.e, the session: trial number interaction effect) was not significantly different from zero ($b = -0.00057, SE = 0.00092, p = 0.53$), suggesting surprise remained relatively stable after the reversal. These results are reported in page 5 of the supplement.

In addition, we briefly touch upon this in the discussion to clarify how surprise dynamics relate to learning categorical state-transitions in our task (Discussion, Page 15, lines 277-283):

“We further linked the model-free and model-based results by means of a supplemental analysis where computationally-estimated surprise was matched against the trial-by-trial behavioural data (Supplement). We found that the trial-by-trial evolution surprise aligned with our model-free

behavioural findings, with surprise decreasing as the task progressed. At the same time, there was no significant interaction between surprise and the post-reversal session, suggesting that participants either adapted to the reversal or that any effects diminished over time.”

[R1.12]

Lines 442-445: Generalised HGF equations: The free model parameters ω is not defined or included in Equation 3. To avoid confusion, the authors could indicate that this parameter appears in Supp Materials. On the other hand, given that this parameter is important for understanding the results, it would be important to briefly define it or introduce it in the equations of the main manuscript.

Thank you for this suggestion to improve clarity. In our revised methods section, we have now explicitly included an explanation of the ω free parameter along with the corresponding equation. Please also see our response to your earlier comment regarding the ω free parameter for a more detailed description of the revisions made.

[R1.13]

line 465: The current recommendation for Gelman-Rubin \hat{r} statistic is 1.01, not 1.1. Which values did the authors obtain?

Vehtari A, Gelman A, Simpson D, Carpenter B, Bürkner PC. Rank-normalization, folding, and localization: An improved \hat{r} for assessing convergence of MCMC (with discussion). Bayesian analysis. 2021 Jun;16(2):667-718.

Thank you for pointing this out. We confirm that all Gelman-Rubin \hat{r} statistic values were less than 1.01. We have revised this sentence and added the above reference to the current recommendation of 1.01.

The revised line now reads as follows (Methods, page 25, lines 582-584):

“Model convergence was confirmed through manual inspection (Supplement, Fig S1) and Gelman-Rubin \hat{r} statistic < 1.01 ^{87,88}.”

For further reference, the \hat{r} statistic values can be found along with the HGF results in the file “*hgf_posteriors_all_participants.csv*”, located in the folder “*hgf_results_summary*” in the Figshare data repository: <https://figshare.com/s/7a99093bbacc7b905463>

Minor:

Some small typos include

[R1.14] adding stop sign before closing “” statements. → line 31 refer to as “uncertainty fingerprints.”

[R1.15] missing en-dashes (some en-dashes are followed by a dash at the closing clarification): lines 47-48: “Minimising these prediction errors – as is central to predictive coding models 31 - may be”

[R1.16] l310: pparticipants

[R1.17] Supp Materials: p6: “agonistic” should be perhaps agnostic?

Thank you for pointing the above typographical errors out. We have corrected them.

[R1.18] l175: , β_4 . Post – error is provided as (0.10, 0.07), but this may be the reverse? (0.07, 0.10)?

We have double-checked this and can confirm it is correct. The values were reported in the order of mean and SD. To avoid confusion, we have revised the phrasing for clarity:

“At the group level, the mean (SD) estimates for each regressor of the response model were ..”

Reviewer #1 (Remarks on code availability):

The Rmd files contain enough annotations for readers to understand the analyses steps. These could be useful for future researchers.

I only checked the Matlab files briefly, but they seem in line with my knowledge of Psychtoolbox. I did not run the code, though.

README has only minimal information and directs the reader to the paper. It would be useful to mention which aspects of the analysis are included in the Rmd files. While this is obvious once one checks the Rmd files, it could be useful to have an outline at this top hierarchical level represented by the README file.

Thanks

We have updated the README file to include a more detailed description of how to run the Matlab task code and provided a clear outline of the analysis steps covered in the Rmd files. The latter description now provides a top level overview to guide users through data preprocessing, behavioural analyses, and figure generation. We believe this improves the usability of the repository: github.com/naziajassim/neurochem_markers_uncertainty

Reviewer #2 (Remarks to the Author):

This is a really interesting paper testing out an exciting concept: linking computational measures of uncertainty to neurochemical data. I was very pleased to review it! The main results are the use of a novel version of the HGF, seeing which elements slow RTs on the SRT task (in this case, just reversal), and linking the results from this to Glx levels in M1, measured using 7T MRS. It is also useful to see the lack of replication of the trait-anxiety/volatility finding, which will add to computational psychiatry literature. However, I had a few concerns.

We thank the reviewer for their thoughtful and encouraging feedback. Below, we address each of the reviewer’s concerns in detail.

[R2.01]

Firstly, I was not sure to what extent the specific tests were planned. For instance, I don't know what 'high-level delta' is, at which timepoint it is calculated, or why that was chosen as a variable to relate to Glx level. The argument of the rest of the paper is that you get an 'uncertainty fingerprint', but 'high-level delta' isn't part of that.

Thank you for raising this point. We note that the description of the high-level delta requires more context. To clarify, the high-level delta refers to the prediction error (δ) at the highest continuous level of the HGF hierarchy; specifically, the top-level node tracking beliefs about transition probabilities. This reflects the magnitude of belief updating in response to unexpected changes in transition probabilities, essentially capturing individual sensitivity to environmental volatility. It is calculated on a trial-by-trial basis and was averaged across the task for each participant and included in the MRS analysis. We selected this quantity for our analysis relating to Glx levels because it captures individual sensitivity to unexpected environmental change – a process that may involve glutamatergic signalling. While this variable is not part of the response model-derived regressors, it reflects a distinct and theoretically relevant mechanism of uncertainty processing that complements the main variables.

We have now added this sentence to the results section (Page 12, lines 203-206) prior to discussing the results of the correlation analysis:

“We examined prediction errors (δ) at the highest continuous level of the HGF hierarchy; specifically, the top-level node tracking beliefs about transition probabilities, which reflects the magnitude of belief updating in response to unexpected changes in transition probabilities.”

You may also be interested in our response to Reviewer #3 [R3.05] about our rationale for choosing to focus on prediction errors over the other computational estimates (see page 21 of this response document).

[R2.02]

I also felt that the 'uncertainty fingerprints' that are derived are not really used or discussed - I think these are a really important contribution. Similarly, other parts of the introduction emphasise certain ideas (e.g. non-binary choices), but this narrative is not continued through the paper.

In our revisions, we have tempered down the “uncertainty fingerprints” terminology as both Reviewer #1 [R1.09] and Reviewer #4 [R4.05] point out, we do not capture a sufficiently extensive or stable set of measures to warrant such framing. As you correctly observe, this concept was not prominently discussed in the manuscript, largely because we aimed to be cautious in our claims. Thus, to strike a more balanced narrative, we omit the use of this specific terminology in our revisions. We believe the strength of this work lies in the integration of behavioural, computational, and neurochemical indices of uncertainty processing to reveal inter-individual differences, and we have therefore used this terminology in our revisions.

We appreciate your suggestion to continue the narrative of non-binary choices/responses throughout the paper. We do believe that our categorical state-transition model captures learning of non-binary outcomes – a novel contribution to understanding how individuals learn complex probabilistic structures beyond binary frameworks. In the revised manuscript, we have clarified this aspect further and ensured that the narrative is consistently maintained.

For example, we have added this to the Discussion (Page 16, Lines 305-310):

“Our key contribution lies in identifying a neurochemical correlate of inter-individual differences in uncertainty processing, specifically within the context of a non-binary, probabilistic learning task. We introduce a novel categorical state-transition extension of the HGF that captures belief updating over multiple discrete outcomes. This model allows us to formally characterise how participants infer transition probabilities in a four-choice environment, moving beyond traditional binary formulations.”

[R2.03]

The introduction in general seems quite technical - more work could be done to further justify the research question and summarise extant literature.

Thank you for this helpful suggestion. During the course of the revisions, we have made several changes to the introduction. Specifically, we have added context on the constructs of uncertainty and volatility, elaborated on their relevance to cognition and mental health, and clarified the role of trait anxiety. We have clarified that motor responses serve as behavioural readouts of implicit learning of the probabilistic sequences and have reduced the discussion of sensorimotor learning to avoid potential confusion. In addition, we have simplified the description of our computational model to enhance accessibility and better situate the research question within the existing literature. Please see the yellow highlighted text throughout the introduction in the revised manuscript.

[R2.04]

My understanding was that at 7T, glutamate and glutamine can be resolved separately (e.g. Lally et al. 2016). Was this not possible using this sequence?

Thank you for raising this important point, which was also raised by Reviewer #3 [R3.07]. You are correct that at 7T, it is in principle possible to resolve glutamate (Glu) and glutamine (Gln) separately with appropriate acquisition and modelling techniques. In our study, we used a short-echo semi-LASER sequence (TR/TE = 5000/26 ms), which offers strong signal-to-noise (SNR) characteristics and robust metabolite quantification.

While separate estimates of Glu and Gln were obtained, we found notable quantification uncertainty in both. We applied quality control criteria based on Cramér-Rao Lower Bounds (CRLBs), excluding values more than 2 standard deviations from the group median, consistent with Near et al. (2013)'s recommendation to avoid fixed CRLB thresholds. Mean CRLBs were 22.3% for Glu and 22.2% for Gln, with ~12% of Gln estimates and ~17% of Glu estimates exceeding 30% CRLB, and some individuals exceeding 40–50%. While not extreme, this level of uncertainty raises concerns about the reliability of interpreting individual Glu or Gln measures. Furthermore, Glu and Gln concentrations were weakly correlated ($r = 0.22$, $p = 0.23$) in our data. Although the correlation is not significant, given that Glu and Gln are biologically linked and arise from overlapping spectral signals, we chose to question the separability and biological specificity of the two signals.

Recent work by Bell et al. (2025) further highlights the complexities involved in separating overlapping metabolites such as glutamate and glutamine at 7T. Their study compared several MRS acquisition sequences (STEAM-8, sLASER-34, and sLASER-105), and assessed their ability to resolve Glu, Gln, GABA, and myo-inositol based on metrics such as CRLBs and SNR. They demonstrated that despite improvements in spectral resolution at 7T, the effectiveness of separating these metabolites is highly dependent on the choice of sequence, voxel placement, and the brain region studied. These findings align with our observation of relatively high uncertainty in Glu and Gln estimates in our motor cortex voxel, supporting our decision to focus on the Glx complex as a more robust and interpretable measure of excitatory neurotransmitter levels.

As we believe yours may be a common question for readers interested in MRS methodology, we have added a note in the Discussion to cover our rationale for focusing on the combined Glx signal rather than individual glutamate measures (Discussion, Page 17-18, lines 349-360):

“Although 7T in principle offers improved spectral resolution, we chose to focus on Glx, as our data showed notable quantification uncertainty for glutamate and glutamine when estimated separately, and the two metabolites were weakly correlated. This is consistent with recent work emphasizing the challenges of reliably separating these metabolites at 7T, which depend heavily on the choice of sequence, voxel placement, and the brain region studied⁶⁰. Focusing on Glx provides a more robust and conservative estimate of excitatory neurotransmitter levels, reducing the risk of overinterpreting potentially noisy or overlapping signals.”

- Near J, Harris AD, Juchem C, Kreis R, Marjańska M, Öz G, et al. Preprocessing, analysis and quantification in single-voxel magnetic resonance spectroscopy: experts' consensus recommendations. NMR Biomed . 2021;34(5):e4257.
- Bell TK, Goerzen D, Near J, Harris AD. Examination of methods to separate overlapping metabolites at 7T. Magn Reson Med. 2025 Feb;93(2):470-480. doi: 10.1002/mrm.30293.

Minor details

[R2.05]

The description of the HGF is generally very clear, but it would be good to know how many higher-order nodes there are and how this is determined (the figure just shows 'a multiplicity' of them).

Thank you for appreciating our description of the HGF. We acknowledge the lack of clarity about the number of higher-order nodes. There are 16 higher-order continuous nodes in our implementation of the categorical state-transition HGF. This number arises because the task involves four possible stimulus categories, and the model tracks transitions between each possible pair of categories. This results in 4 (source categories) × 4 (target categories) = 16 possible transitions, each of which is associated with a dedicated higher-order node $x_{i,j}$. These nodes encode dynamic beliefs about the tendency to transition from category i to category j over time, and together they form the basis for estimating evolving transition probabilities during learning. As it is not possible to clearly visualise all 16

nodes, we use dotted lines in the schematic of the model (Fig. 3a) to indicate their multiplicity.

In the revised manuscript, we explicitly mention the number of nodes and how they were determined:

Figure 3 caption:

“Dotted lines indicate the multiplicity of nodes. In our implementation, there are 16 higher-order nodes, one for each possible transition between four categories (4×4).”

Methods, page 23, lines 524-525:

“There are 16 higher-order continuous nodes, each representing a possible transition between the four categories.”

[R2.06]

Why is subject ID displayed in figure 4's correlogram? This is a little confusing to me.

We thank you for bringing this to our attention. Subject ID was inadvertently included in the original correlation matrix, which understandably caused confusion in the figure. We have now excluded Subject ID and have updated Figure 4 accordingly. The matrix was included as supplementary information for transparency in MRS reporting by showing the relationships among all quantified (non-tissue-corrected) metabolites, and to highlight Glx and GABA, which we focus on in our main and control analyses. We can confirm that the correlation between Glx and GABA remains visually unchanged in the revised figure:

Left panel: Original correlogram. Right panel: Revised (corrected) correlogram.

[R2.07]

There are some typos in the methods section - e.g. 'pparticipants' and saying that sequence B is both the most and least probable in session 2, 'assessed BY means...' on line 391.

Thank you for pointing this out. We confirm that we have fixed these.

[R2.08]

I was not able to access the dataset through the figshare link.

We apologise for the issue with accessing the dataset. Please note that the Figshare DOI referenced in the manuscript's data availability statement is not yet publicly accessible and will be made available upon publication. However, the dataset should be accessible to reviewers via this private link: <https://figshare.com/s/7a99093bbacc7b905463>.

[R2.09]

Figure S1 is a little unclear - what are the error bars in plot A? Equally, Figure S2 is fairly grainy - and in the description, you say that surprise increases in the middle and late parts of the post-reversal session, but the graph (which is binned differently) doesn't reflect this.

Thank you for pointing out the lack of clarity in these figures and their descriptions so that we can make the appropriate revisions.

Regarding Fig S1: Panel A of Figure S1 indicates the prior (green) and posterior (orange) distributions, while the surrounding shaded areas represent the respective distributions themselves. The overlaid bars indicate the 50% and 80% credible intervals (not error bars), which as is standard in Bayesian inference, summarise the central range of the distributions. We have clarified this in the revised figure caption as follows:

“A) Prior (green) and posterior (orange) probability distributions for each parameter. Overlapping distributions show how the posterior is informed by the data relative to the prior. Vertical bars represent 50% and 80% credible intervals of the distribution. Circles indicate the medians.”

Regarding Fig S2 and the corresponding analysis: In our revisions, we have changed this analysis based on feedback from Reviewer #1 [R1.06, R1.11] encouraging us to focus on trial number rather than learning stages in our linear mixed-effects models. Specifically, our revised LME analysis models surprise as a function of trial number, session, and their interaction, including random intercepts and slopes for trial number by participant. Results showed that surprise significantly decreased across trials during the pre-reversal session, indicating learning over time. However, there was no significant change in surprise across trials during the post-reversal session, suggesting surprise remained relatively stable after the reversal. Please also see our responses to Reviewer #1 [R1.06, R1.11] for further details about this revised LME analysis.

Due to the large number of trials in the task, it is challenging to directly visualise trial-by-trial learning. For visualisation purposes only, we have binned the data into blocks of 120 trials, which allows for a clearer depiction of learning trajectories across the task.

Based on your comment, we have improved the resolution of Figures S1 and S2 in the revised manuscript.

Reviewer #2 (Remarks on code availability):

The code accessible in the github seems to use pre-cleaned data, it would be good if the data cleaning code was available too. Equally, the code to launch the models wasn't included.

Thank you for pointing this out. We confirm that the data cleaning and preparation steps for the model-free behavioural analyses are included in the Rmd scripts, which are available here:

https://github.com/naziajassim/neurochem_markers_uncertainty

The HGF modelling and fitting scripts, including model launching and fitting workflows, are provided in a separate repository:

https://github.com/naziajassim/hgf_srt

In addition, we have updated the README files in both repositories to include more details on how to run the task and reproduce the analysis, modelling, and figures reported in the manuscript.

The data on figshare was not available, so I was not able to see whether the code would be a usable resource.

We apologise for the earlier issue regarding data access on Figshare. Please note that the Figshare DOI cited in the manuscript's Data Availability statement is not yet publicly accessible but will be made available upon publication. In the meantime, the data can be accessed by reviewers via the following private link:

<https://figshare.com/s/7a99093bbacc7b905463>

Reviewer #3 (Remarks to the Author):

I enjoyed reading this paper by Jassim and colleagues, in which they report associations between probabilistic reinforcement learning processes and glutamate/glutamine assessed using magnetic resonance spectroscopy. They demonstrate that prediction errors and beliefs about environmental volatility during a motor learning task are linked to Glx levels (a compound measure of glutamate and glutamine) in the motor cortex, suggesting that glutamatergic neurotransmission in this region may be involved in motor learning.

The study is innovative and the methods appear robust, while the paper is well-written (save for some points of clarification that may assist the reader). Below I note some suggestions that could be incorporated into a revision.

We greatly appreciate the reviewer's insightful and supportive commentary on our work. We are pleased to address the reviewer's point by point comments in detail below.

[R3.01]

The computational modelling is somewhat difficult to understand as it seems to diverge somewhat from typical applications of computational modelling to probabilistic learning tasks. First, it is unclear why the model was fit to log RT rather than to responses directly, which would presumably provide a clearer readout of how participants are learning about the probability of each item in the sequence.

Thank you for this point. We agree that many computational models of probabilistic learning are fit directly to participants' choices or discrete responses, particularly in tasks involving rewards/punishments that require overt decisions or action selection. However, in our case, the task is a non-instructed, implicit probabilistic sequence learning paradigm, where participants respond to each item in a pre-specified manner (via a button press), and RT, rather than choice, serves as the primary behavioural measure of learning. There are no probabilistic decisions to be made per se (as in common in probabilistic reward learning tasks), but rather, learning is reflected in facilitated RTs to high probability stimuli. Specifically, the probabilistic serial reaction time (SRT) task captures this implicit learning dynamic, with RTs serving as the core behavioural index of learning (Nissen & Bullemer, 1987; Kaufman et al., 2010; Jiménez & Méndez, 1999).

As other readers may have the same question, we have now prefaced the description of the HGF response model with a brief justification for using log RT (Methods, Page 24, Lines 561-564):

“In the probabilistic SRT task, response speed, rather than choice, serves as the primary measure reflecting participants' implicit learning of stimulus probabilities^{3,73,74}. Accordingly, we modelled the log-transformed RTs using a linear regression that ...”

- Nissen, M. J., & Bullemer, P. (1987). Attentional requirements of learning: Evidence from performance measures. *Cognitive Psychology*, 19(1), 1–32. [https://doi.org/10.1016/0010-0285\(87\)90002-8](https://doi.org/10.1016/0010-0285(87)90002-8)
- Kaufman SB, DeYoung CG, Gray JR, Jiménez L, Brown J, Mackintosh N. Implicit learning as an ability. *Cognition*. 2010 Sep;116(3):321–40.
- Jiménez, L., & Méndez, C. (1999). Which attention is needed for implicit sequence learning? *Journal of Experimental Psychology: Learning, Memory, and Cognition*, 25(1), 236–259. <https://doi.org/10.1037/0278-7393.25.1.236>

[R3.02]

Second, the regression approach to model fitting seems like a somewhat indirect way to infer how participants are estimating uncertainty. Ordinarily we might expect the model itself to have free parameters (e.g., learning rates) that can capture different learning profiles depending on their value; here it seems that the learning model itself is fairly rigid (aside from the expectation of volatility) and instead it is the extent to which this rigid model predicts participants' behaviour that is being estimated. Does this affect the interpretation of the results?

Thank you for this insightful observation. While the HGF model we used does not include an explicit learning rate parameter, the learning rate is dynamically determined by the agent's beliefs about volatility. These govern the rate at which beliefs are updated and thus

serve as implicit, dynamic learning rates. It is common in the HGF framework to use a separate regression model to map belief trajectories onto behaviour (e.g., Lawson et al., 2017; Marshall et al., 2016, Diaconescu et al., 2014; Iglesias et al., 2013). This approach, while not as mechanistic as fully generative models of behaviour, allows for testing targeted hypotheses about how specific computational variables (e.g., volatility beliefs, different types uncertainty) relate to observed behaviour.

We agree that a more principled, generative response model could yield richer insights. Our current approach enables testing specific hypotheses about the influence of volatility beliefs, surprise, different types of uncertainty, and specific task-related parameters on behaviour, but we acknowledge it limits the mechanistic interpretation of learning processes.

Based on your comment, we have added the following to the discussion section (Page 15, lines 268-276):

“While the HGF model used here does not include an explicit learning rate parameter, learning rates are dynamically governed by participants’ inferred volatility beliefs, which shape the rate at which beliefs are updated. This dynamic updating mechanism effectively serves as an implicit, trial-by-trial learning rate. Consistent with prior work employing the HGF framework^{26,27,52}, our approach uses a separate regression model to relate these computational variables to behaviour. Although less mechanistic than fully generative models, this allows us to test targeted hypotheses about how specific computational signals and task-related parameters influence response time dynamics. Future work could extend these insights by incorporating more mechanistic response models.”

- Marshall, L., Mathys, C., Ruge, D., de Berker, A. O., Dayan, P., Stephan, K. E., & Bestmann, S. (2016). Pharmacological Fingerprints of Contextual Uncertainty. *PLoS Biology*, 14(11), e1002575. <https://doi.org/10.1371/journal.pbio.1002575>
- Diaconescu, A. O., Mathys, C., Weber, L. A. E., Daunizeau, J., Kasper, L., Lomakina, E. I., ... & Stephan, K. E. (2014). Inferring on the intentions of others by hierarchical Bayesian learning. *PLoS Computational Biology*, 10(9), e1003810. <https://doi.org/10.1371/journal.pcbi.1003810>
- Iglesias, S., Mathys, C., Brodersen, K. H., Kasper, L., Piccirelli, M., den Ouden, H. E., & Stephan, K. E. (2013). Hierarchical Prediction Errors in Midbrain and Basal Forebrain during Sensory Learning. *Neuron*, 80(2), 519–530. <https://doi.org/10.1016/j.neuron.2013.09.009>
- Lawson, R. P., Mathys, C., & Rees, G. (2017). Adults with autism overestimate the volatility of the sensory environment. *Nature Neuroscience*, 20(9), 1293–1299. <https://doi.org/10.1038/nn.4579>

[R3.03]

For the comparison models reported in the supplementary material, how were these fit to the data? Did this use a similar regression-style approach?

Thank you for this question. The model comparison was conducted primarily as a validation step to evaluate the appropriateness of alternative RT-based learning models and to demonstrate that the categorical state-transition HGF offered the most explanatory power for our dataset. All three models, including the Rescorla-Wagner and EWA comparison models, were implemented and fit within a hierarchical Bayesian framework.

While these models do not use the same regression-based approach as the HGF, they each estimate trial-by-trial learning signals (e.g., prediction errors or experience-weighted values) that inform response tendencies. These signals were linked to reaction times via softmax-based mappings, allowing for likelihood-based comparison using PSIS-LOO.

Based on your comment, we have added the following clarification to the Model Comparison section in the Supplement (page 3):

“While Models 2 and 3 do not use the same regression-based approach as Model 1, all three the comparison models estimate trial-by-trial learning signals (e.g., prediction errors or experience-weighted values) that inform response tendencies”

[R3.04]

The concept of volatility could be introduced a little better within the context of the present task. Since it does not feature a clear volatility manipulation (aside from the reversal), it can be a little hard to understand why we might be expecting participants to track volatility at all.

Thank you for this comment, which echoes a similar point raised by Reviewer #4 [R4.04]. We have added a paragraph to the introduction (page 3, lines 29-36) to better introduce the concept of volatility:

“Emerging evidence suggests that different types of uncertainty, such as expected uncertainty (known variability of outcomes), unexpected uncertainty (sudden deviations from expected patterns), and volatility (the changeability of the environment over time), may be represented by partially distinct neural mechanisms and computational processes^{7,8,10,25,26}. Volatility, a higher-order form of uncertainty, reflects individuals’ inferred uncertainty about potential changes in the environment. In an uncertain environment, individuals track volatility and adjust their learning accordingly, weighting new information more heavily when volatility is perceived to be high, and more conservatively when volatility is low^{5,10,25-27}.”

As for your second point, we agree that volatility is not explicitly manipulated in the task beyond the reversal; however, volatility reflects participants’ inferred uncertainty about potential changes in the environment’s structure and is a core latent quantity estimated by the Hierarchical Gaussian Filter (HGF). Even in relatively stable tasks, the HGF allows us to model individual differences in how strongly participants expect the environment to change (i.e., volatility), which may shape their learning and surprise responses following the reversal. The parameter w is crucial because it encodes an individual’s estimate of environmental volatility, allowing the HGF to dynamically adjust learning rates based on changing uncertainty; this sets it apart from traditional learning models with fixed update rates.

We have now added a brief justification for the inclusion of volatility beliefs in our analyses (Discussion, Page 17, lines 325–327), where we address the relevant findings:

“Our model captures individual differences in inferred environmental volatility, offering insight into how participants internally model the stability or instability of the probabilistic structure, even in the absence of explicit volatility manipulations.”

[R3.05]

Is there a reason for using average prediction errors in the MRS analyses rather than surprise, given that surprise seems more relevant to the Bayesian updating process?

Thank you for raising this point. We agree that surprise is a central quantity in Bayesian updating frameworks. However, we focused on average prediction errors (PEs) in our MRS analyses for a few reasons. First, prediction errors are a core concept in both Bayesian and reinforcement learning models and have a well-established neurobiological interpretation in predictive coding accounts, where they are thought to drive the minimisation of surprise (Friston, 2005; Feldman & Friston, 2010). Prediction error signals have been repeatedly linked to neuromodulatory and neurotransmitter systems (e.g., O'Doherty et al., 2003; Iglesias et al., 2013), making them a theoretically and empirically grounded target for MRS-based analyses. Furthermore, we chose prediction errors over surprise specifically to aid interpretability for a broader audience familiar with reinforcement learning and error-driven learning models (Sutton & Barto, 2018).

Thus, while surprise is indeed conceptually distinct and relevant, especially in the context of belief updating, prediction error offers a more direct link to prior neuroscience findings. That said, we acknowledge that surprise is a complementary and informative computational quantity, and we appreciate the opportunity to clarify our reasoning.

- O'Doherty, J. P., Dayan, P., Friston, K., Critchley, H., & Dolan, R. J. (2003). Temporal difference models and reward-related learning in the human brain. *Neuron*, 38(2), 329–337.
- Friston, K. (2005). A theory of cortical responses. *Philosophical Transactions of the Royal Society B*, 360(1456), 815–836.
- Feldman, H., & Friston, K. J. (2010). Attention, uncertainty, and free-energy. *Frontiers in Human Neuroscience*, 4, 215.
- Iglesias, S., Mathys, C., Brodersen, K. H., Kasper, L., Piccirelli, M., den Ouden, H. E., & Stephan, K. E. (2013). Hierarchical prediction errors in midbrain and septum during social learning. *Neuron*, 80(3), 519–530.
- Sutton, R. S., & Barto, A. G. (2018). *Reinforcement Learning: An Introduction* (2nd ed.). MIT Press.

[R3.06]

Could the PE-Glx effect be confounded by performance? I.e., worse performers would see greater prediction errors, and performance itself may be related to Glx.

Thank you for raising this interesting point. To address this possibility, we examined the relationship between overall accuracy and M1 Glx concentrations. This indicated no significant correlation between M1 Glx levels and task performance accuracy ($r = 0.11$, $t(68) = 0.94$, $p = 0.35$, 95% CI $[-0.13, 0.34]$). This rules out the possibility that Glx–prediction error association may reflect a performance-related confound and supports the interpretation that M1 Glx is specifically related to computational learning signals.

We have added the above minor confirmatory analysis to page 7 of the Supplement.

[R3.07]

Given MRS was performed at 7T, it should be feasible to separate glutamate from glutamine rather than using the combined Glx measure, was this attempted?

We appreciate you raising this important point, which was also noted by Reviewer #2 [R2.04]. We did attempt to separately quantify glutamate and glutamine using our 7T MRS data; however, due to notable quantification uncertainty in both estimates, we opted to focus on the more reliable combined Glx measure. Please see our detailed response to Reviewer #2 [R2.04] on page 13 of this document for further explanation.

[R3.08]

How far apart in time were the scan and the behavioural testing?

Due to scheduling constraints, the scan and behavioural testing were conducted on separate days but always within the same week. The behavioural testing sessions were always prior to the scan. We have added the following sentence to the Methods section (Page 20, lines 437-438) for added clarity:

“The scan took place on a separate day from the behavioural testing session but was conducted within the same week.”

[R3.09]

Some of the phrasing could be toned down a little to avoid overstating the results. For example p16 “M1 Glx plays a critical role” – it is hard to say from a correlational study like this whether it really is critical.

We agree that this could be more cautiously framed and have changed the sentence as follows:

“...revealing M1 Glx as an important neurochemical correlate of volatility beliefs and prediction errors during implicit learning.”

[R3.10]

A technical question: for a task that involves estimating probabilities, it seems like using multiple Gaussian distributions to represent these probabilities is a little awkward. Is there any way the model could be respecified to use a Dirichlet distribution that would more easily capture these values? I don't necessarily expect the authors to do this, it's more a point of curiosity.

We appreciate your insightful observation regarding the potential mismatch between using Gaussian-distributed variables and the task's probabilistic nature. Although the model includes Gaussian dynamics, these are applied in a principled way that respects the nature of the task and supports structured learning over categorical outcomes.

To clarify, in our model, we do indeed represent the categorical transition probabilities using a Dirichlet distribution, which is a natural choice for modelling probabilities over

discrete outcomes. However, it is crucial to note that we filter the sufficient statistics of this Dirichlet distribution using the HGF nodes. Note that the HGF can be applied to exponential family distributions, including the Dirichlet, by operating on their sufficient statistics (Mathys & Weber, 2020). In practice, this means that while the predictions themselves are in a categorical space (captured by the Dirichlet mean), the underlying learning about how those probabilities evolve over time is governed by Gaussian random walks, consistent with the HGF formalism. This allows us to retain the Bayesian belief updating properties of the HGF while respecting the categorical structure of the task.

Thus, while at first glance it might seem that a purely Gaussian structure is an awkward fit for a probabilistic task, in our case, the Gaussian nodes are used appropriately to track changes in probabilistic beliefs via the sufficient statistics of a Dirichlet distribution.

- Mathys, C., Weber, L. (2020). Hierarchical Gaussian Filtering of Sufficient Statistic Time Series for Active Inference. In: Verbelen, T., Lanillos, P., Buckley, C.L., De Boom, C. (eds) Active Inference. IWAI 2020. Communications in Computer and Information Science, vol 1326. Springer, Cham. https://doi.org/10.1007/978-3-030-64919-7_7

[R3.11]

Minor point: p9, it may be worth rephrasing “these beliefs are updated faster” as it can be easy to misread “faster” as meaning “faster in time” within the context of reaction times and post-error slowing.

Thank you for pointing out the potential for confusion here. We have changed the phrasing to as follows:

“Belief updates are larger in magnitude when..”

Reviewer #4 (Remarks to the Author):

The paper titled “Neurochemical markers of uncertainty processing in humans” by Jassim and colleagues presents a unique investigation linking computational modelling and MR spectroscopy. Specifically, the authors use a novel variant of the Hierarchical Gaussian Filter to estimate volatility in a serial reaction time task and link it to behaviour as well as glutamate/glutamine levels in the motor cortex. Additionally, they associate individual differences in trait anxiety to acceleration of reaction times after reversal.

This investigation constitutes a well-executed and technically sophisticated study generating novel insights into the neurochemical mechanisms of uncertainty. Below I present a list of questions and suggestions - I would be happy to recommend the paper for publication if these can be addressed. Though, ultimately, I leave it up to the authors which they choose to incorporate in their paper.

We thank the reviewer for their favourable appraisal of our study. It is encouraging to know that the conceptual approach and findings were well received. We have carefully considered all the questions and suggestions raised and have addressed them below.

[R4.01]

Title - I think the title should more closely reflect the anatomical specificity of the analyses, i.e. it should mention i) motor cortex and ii) glutamate (Glx). As it stands, it gives the impression that a broad range of neurochemical systems and regions has been investigated.

Thank you for your helpful suggestion. We revised the title from the broader “Neurochemical markers of uncertainty processing” to “Computational signatures of uncertainty processing are reflected in motor cortex excitatory neurochemistry” to explicitly specify the anatomical focus on the motor cortex and the neurochemical emphasis on excitation. We do not mention glutamate here because our MRS data reflects the combined glutamate + glutamine (Glx) signal rather than glutamate alone. Additionally, “Glx” is a technical term mainly familiar to MRS researchers and might not be accessible to a broader audience. We believe this wording balances scientific precision with clarity and ensures the title resonates with the broad interdisciplinary readership of the journal.

[R4.02]

It is unclear why M1 was selected as the target region for MRS. Other regions, such as the OFC, ACC, striatum etc (e.g. Behrens et al., 2007; Muller et al., 2019) have been more consistently linked to uncertainty, and they would be more likely a priori candidates.

Thank you for raising this point. We agree that “higher-order” regions such as the OFC, ACC, and striatum are prominently associated with uncertainty processing, particularly in paradigms involving explicit outcome monitoring or value-based decision-making. In contrast, our study employed an *implicit* learning paradigm in which participants were not consciously aware of the probabilistic structure and were not required to explicitly evaluate or act upon uncertainty. Under such conditions, we hypothesized that uncertainty may be processed at a more automatic, sensory level, with sensorimotor regions tracking environmental regularities through accumulated experience. We selected the primary motor cortex (M1) as our MRS target based on several recent lines of evidence:

- **M1 demonstrates neurochemical plasticity in response to motor learning demands, even without explicit awareness (Kolasinski et al., 2019).**
- **Visual and feedback-related uncertainty exert distinct effects on motor cortical activity during reach planning, indicating that different forms of uncertainty are processed within motor cortex (Amann et al., 2024).**
- **M1 activity is biased by stimulus history, supporting its role in integrating environmental regularities (Braun & Donner, 2023).**
- **Motor regions can encode decision variables such as value and confidence, even in tasks lacking explicit choice demands (Aquino et al., 2023).**
- **Theoretical frameworks highlight the role of sensorimotor systems in contextual inference and probabilistic learning, especially under uncertainty (Heald et al., 2021).**

Taken together, these findings support the view that M1 is well-suited to reflect experience-driven learning and uncertainty tracking at a sensorimotor level, making it a justifiable target for MRS in our study.

Additionally, the Serial Reaction Time Task used in our study has been repeatedly shown to involve M1 during both online learning and consolidation phases of implicit sequence learning (e.g., Honda et al., 1998; Hardwick et al., 2013; Kolasinski et al., 2019). These findings provide further justification for our targeting of M1 in the context of a task that taps into motor adaptation and learning under uncertainty.

Based on your comment, we have added the following paragraph to the Discussion section (Page 17, lines 361-373) which explains our rationale for choosing M1 as our MRS target:

“Prior studies of uncertainty processing have implicated frontal and subcortical regions such as the orbitofrontal cortex, anterior cingulate cortex, and striatum, particularly in value-based or explicit outcome monitoring paradigms^{5,8,10}. In contrast, our study employed an implicit learning paradigm in which participants were neither consciously aware of the underlying probabilistic structure nor required to explicitly make choices or respond to uncertainty. Given this design, we hypothesised that uncertainty is processed in a more automatic, sensory fashion, with sensorimotor regions tracking environmental regularities through accumulated experience. The primary motor cortex (M1) is a well-suited MRS target given its established role in implicit motor learning and sensorimotor plasticity. Prior work has shown that M1 exhibits neurochemical changes, such as GABA modulation, in response to learning demands even in the absence of explicit awareness⁶³. Moreover, M1 has been implicated in tracking environmental regularities and processing feedback-related and sensory uncertainty^{64–66}, suggesting it may play a role in implicit learning under uncertainty.”

- Amann, L.K., Casasnovas, V. & Gail, A. Visual target and task-critical feedback uncertainty impair different stages of reach planning in motor cortex. *Nat Commun* 16, 3372 (2025). <https://doi.org/10.1038/s41467-025-58738-x>
- Aquino TG, Cockburn J, Mamelak AN, Rutishauser U, O'Doherty JP. Neurons in human pre-supplementary motor area encode key computations for value-based choice. *Nat Hum Behav.* 2023 Jun;7(6):970-985. doi: 10.1038/s41562-023-01548-2. Epub 2023 Mar 23. PMID: 36959327; PMCID: PMC10330469.
- Behrens, T. E., Woolrich, M. W., Walton, M. E., & Rushworth, M. F. (2007). Learning the value of information in an uncertain world. *Nature Neuroscience*, 10(9), 1214–1221. <https://doi.org/10.1038/nn1954>
- Braun A, Donner TH (2023) Adaptive biasing of action-selective cortical build-up activity by stimulus history *eLife* 12:RP86740 <https://doi.org/10.7554/eLife.86740.3>
- Hardwick, R. M., Rottschy, C., Miall, R. C., & Eickhoff, S. B. (2013). A quantitative meta-analysis and review of motor learning in the human brain. *NeuroImage*, 67, 283–297. <https://doi.org/10.1016/j.neuroimage.2012.11.020>
- Honda, M., Deiber, M. P., Ibáñez, V., Pascual-Leone, A., Zhuang, P., & Hallett, M. (1998). Dynamic cortical involvement in implicit and explicit motor sequence learning: A PET study. *Brain*, 121(11), 2159–2173. <https://doi.org/10.1093/brain/121.11.2159>
- Kolasinski, J., Hinson, E. L., Divanbeighi Zand, A. P., Rizov, A., Emir, U. E., & Stagg, C. J. (2019). The dynamics of cortical GABA in human motor learning. *The Journal of Physiology*, 597(1), 271–282. <https://doi.org/10.1113/JP276626>

- Müller, F., van der Marel, K., & van den Heuvel, O. A. (2019). Neural correlates of uncertainty in decision making: A meta-analysis. *Neuroscience & Biobehavioral Reviews*, 107, 145–156. <https://doi.org/10.1016/j.neubiorev.2019.09.003>
- Heald, J. B., Lengyel, M., & Wolpert, D. M. (2021). Contextual inference underlies the learning of sensorimotor repertoires. *Nature*, 600(7887), 489–493. <https://doi.org/10.1038/s41586-021-04120-1>

[R4.03]

When discussing the Glx concentrations in relation to high-level PE and volatility beliefs, could the authors add whether such PEs and volatility estimates correlate? This is to clarify whether the two are related or whether the glutamate levels are related to both independently. If they relate, this could also be added to the discussion.

Thank you for this valuable suggestion. We indeed expect high-level prediction errors and volatility beliefs to correlate. Prediction errors index the magnitude of surprise on trials, while volatility beliefs reflect inferred environmental stability that shapes how prediction errors are integrated over time. In our implementation, ω is the only free parameter of the HGF model, governing belief-updating at high levels. As such, variation in this parameter directly shapes the dynamics of volatility beliefs and their influence on prediction errors. Because ω determines the degree to which the model expects the environment to change, it substantially governs the responsiveness of belief updating, and therefore, the variability of prediction errors. This tight coupling would likely contribute to a strong inverse relationship between these quantities.

Based on your recommendation, we conducted an additional correlation analysis between these two computational quantities. This revealed a strong negative correlation ($r = -0.983$, $p < 2.2 \times 10^{-16}$), indicating that participants with lower volatility beliefs exhibited larger high-level prediction errors. This finding suggests that the relationships observed with Glx levels may reflect their joint influence on both belief stability and error-driven learning under uncertainty. We have added this analysis to page 8 of the Supplement, reference it in the Results, and have integrated it into the Discussion.

Results (Page 12, lines 214-222):

“Notably, in our implementation of the HGF, ω is the sole free parameter and governs belief updating at higher levels of the hierarchy. As such, variation in ω shapes both volatility estimates and their influence on the magnitude of prediction errors (δ). Indeed, a separate correlation analysis confirmed a strong inverse relationship between these quantities (Supplement), indicating that participants with lower inferred volatility exhibited larger high-level prediction errors. Taken together, these findings suggest that the above associations with Glx may reflect its influence on a shared neurocomputational mechanism underlying both volatility beliefs and high-level prediction errors.”

Discussion (Page 17, lines 331-338):

“Furthermore, a supplementary correlation analysis revealed a strong negative relationship between high-level prediction errors and volatility beliefs, indicating that participants who expected lower

environmental volatility showed larger high-level prediction errors (Supplement). In other words, when the environment was perceived as more stable, surprising outcomes had greater computational impact, resulting in larger belief updates. This relationship provides a mechanistic bridge between Glx levels, volatility beliefs, and prediction error dynamics: elevated Glx may lead individuals to expect a more stable environment, thereby amplifying the impact of unexpected outcomes and enhancing the learning signal when predictions are violated.”

[R4.04]

In the introduction, I think it might help readers to have a bit more introduction into the topic of uncertainty. Specifically, what is volatility and unexpected/expected uncertainty and how they are related, following from a nice overview in Sandhu, Xiao & Lawson 2023.

Thank you for this suggestion, which is similar to that of Reviewer #3 [R3.04]. We have added another paragraph to the introduction, in which we introduce the concepts of uncertainty and volatility (Page 3, lines 29-36):

“Emerging evidence suggests that different types of uncertainty, such as expected uncertainty (known variability of outcomes), unexpected uncertainty (sudden deviations from expected patterns), and volatility (the changeability of the environment over time), may be represented by partially distinct neural mechanisms and computational processes^{7,8,10,25,26}. Volatility, a higher-order form of uncertainty, reflects individuals’ inferred uncertainty about potential changes in the environment. In an uncertain environment, individuals track volatility and adjust their learning accordingly, weighting new information more heavily when volatility is perceived to be high, and more conservatively when volatility is low^{5,10,25-27}.”

[R4.05]

On a related point, I would suggest toning down the use of “uncertainty fingerprints” as it gives the impression that an extensive array of uncertainty measures has been used. While several aspects of learning under uncertainty are estimated, this is by no means exhaustive.

We agree with your observation, which was also made by Reviewer #1 [R1.09]. We have omitted the term “uncertainty fingerprints” from the revised manuscript and have replaced it with “individual differences in uncertainty processing”. Please see our detailed response to Reviewer #1 on Page 7 of this document for specific examples of these revisions.

[R4.06]

Task - it is somewhat unclear whether unlikely transitions (0.15) lead to a position within the same sequence or whether they lead to sequence-to-sequence transitions. I was assuming that the transitions are within-sequence, especially given the high accuracy.

We appreciate the opportunity to clarify this point. The unlikely transitions (0.15 probability) do not reflect transitions between Sequence A and Sequence B per se, but rather correspond to less probable second-order transitions within a single unified Markov-chain framework. Specifically, each pair of preceding positions ($t-2$, $t-1$) is

associated with a high-probability (85%) third position (t) and a low-probability (15%) alternative. These transitions are drawn from both predefined sequences, but at any given time during the task, one sequence governs the high-probability transitions. Thus, all transitions occur within the same overarching sequence structure. In other words, participants are not switching between entire sequences on a trial-by-trial basis.

Based on your comment, we have added the following clarification to the task description (Methods, page 19, lines 432-427):

“The position on trial t is determined by the position of the two previous trials, $t-1$ and $t-2$. The two deciding sequences differ in the second-order conditional occurrence, with one occurring 85% of the time and the other 15%⁷⁴⁻⁷⁶ (Fig 1b-d). This means that, although transitions are derived from two predefined sequences (Sequence A and Sequence B), both high and low probability transitions occur within the same unified probabilistic framework. In other words, these transitions do not reflect switches between entire sequences but instead represent more or less frequent transitions within the same overarching Markov structure governing the task.”

[R4.07]

Figure 3c - this point may stem from my lack of understanding of aspects of the model - why are the state transition probabilities roughly equal for the three non-current options [~ 0.3 , ~ 0.3 , ~ 0.3 , 0]? Shouldn't learning lead them to reflect the true transition probabilities [0.85, 0.15, 0, 0]?

Thank you for this clarification question. The pattern observed in Fig. 3c is based on both the i) task structure and ii) the nature of the HGF model:

- i. Fig. 3c reflects the trial-by-trial transition beliefs, and not the second-order transition probabilities of the task. To elaborate, the task employs two embedded Markov chains (Sequences A and B), and transitions depend on the preceding two stimuli (second-order). This creates a set of second-order contexts, each associated with different transition distributions. For many of these contexts, the upcoming stimulus is not uniquely predictable. Thus, even if the global frequency of transitions aligns with 85/15, any single trial-to-trial transition may not.
- ii. Fig. 3c reflects the agent's *beliefs*, and not the ground truth. The HGF doesn't "recover" the true generative structure directly; instead, it forms subjective beliefs based on the input and its own assumptions (e.g., about volatility, priors). Thus the values shown in Fig. 3c represent the model's inferred beliefs about transition probabilities, not the actual task contingencies. The example session displayed shows the model is accumulating evidence about a specific transition. When multiple outcomes remain plausible, the model may assign near-equal probabilities to all allowed transitions (excluding transitions between the same Category, which the model learns is impossible, and thus calculates to be zero). This reflects uncertainty, not incorrect learning.

[R4.08]

Reaction times are often believed to reflect aspects of uncertainty processing. Here, however, RTs do not seem to reflect model-based estimates. While this is made clear in figure 4a, it might also deserve a minor discussion point.

Thank you for highlighting this. We have now briefly discussed this finding in our revised discussion (Page 15, lines 258-264):

“Although response times are often assumed to reflect internal uncertainty estimates, our HGF response model indicated that reaction times were not consistently modulated by belief-based predictors such as expected or unexpected uncertainty. Apart from a reliable post-error slowing effect, the belief-related regressors showed no significant group-level effects, suggesting that reaction times may capture the behavioural consequences of learning, such as post-error slowing, rather than belief updating itself.”

[R4.09]

In the introduction, the selection of TA is not directly justified using past literature.

Thank you for this suggestion. We have added additional background context about the relevance of trait anxiety in the revised Introduction (Page 3, lines 36-40):

“...Individual differences in these computations have been linked to trait anxiety, with early studies suggesting that more anxious individuals may overestimate volatility or update beliefs less adaptively^{12,28}. However, recent large-sample studies have not consistently replicated these findings^{19,29,30}, highlighting the need for further investigation.”

[R4.10]

It is also not clear why more specific constructs were not included (Intolerance of Uncertainty Scale).

We agree that the Intolerance of Uncertainty Scale more specifically targets responses to uncertainty. However, we chose trait anxiety because it has been consistently linked to uncertainty processing in past work and captures a broader emotional vulnerability relevant to mental health. Future studies could benefit from including both measures to tease apart their unique contributions to belief updating.

We now include a brief justification for this choice in the Discussion, where we discuss the findings pertaining to trait anxiety (Page 15, 286-288):

“Although more specific constructs, such as the Intolerance of Uncertainty Scale⁵⁵, may better capture sensitivity to uncertainty, we focused on trait anxiety due to its broader clinical relevance and well-documented links to uncertainty processing. Based on prior findings...”

[R4.11]

At a number of points, the authors refer to a specific signal as a “reversal” e.g. M1 Glx on page 12 (line 187). However, reversal here cannot be dissociated from simple temporal effects. I.e., if

there is an interaction between session and Glx, this could be that Glx levels are more closely associated with RTs in one of the sessions. Please correct this to prevent the wrong impression that the signal is specific contingency reversals.

Thank you for pointing this out. We agree that the observed effects may reflect general temporal dynamics rather than being specific to the contingency reversal. We have now made this explicit in the revised discussion, where we discuss limitations (Page 18, lines 357-360):

“Third, the post-reversal effects on behaviour and belief-updating cannot be conclusively attributed solely to the reversal itself, as they may also reflect general effects of time or task progression. Future work with matched control conditions or counterbalanced designs may help to disambiguate these effects.”

[R4.12]

The “high-level PE” reflects general accuracy of predictions across all positions. I was just wondering if it could be described (perhaps in discussion) in terms of information gain across the entire belief state, in order to provide an intuitive understanding of this aggregate measure. On a related note, were the averaged prediction errors signed or unsigned? In my understanding they should be unsigned otherwise positive and negative PEs cancel each other and result in no “high level PE”.

Thank you for this insightful comment. We agree that providing an intuitive interpretation of the “high-level prediction error” (PE) would help contextualise its role in our analyses. As you note, this quantity reflects the magnitude of belief updating at the highest level of the model, i.e., belief adjustments regarding transition probabilities, and can indeed be interpreted as an index of information gain at the level of abstract beliefs about environmental structure. We have now added this interpretation to the Discussion section (Page 16-17, lines 316-324):

“Learning is driven by prediction errors - the difference between what was expected and what actually occurred^{60,61}. Our model-based findings suggest that higher M1 Glx levels are associated with greater prediction errors, reinforcing the idea that glutamate-related plasticity supports error-driven learning. Importantly, in hierarchical models, the average prediction errors, particularly at higher levels, can also be interpreted in terms of information gain, capturing the extent to which surprising outcomes prompt updates to more abstract beliefs about environmental structure. In this framework, larger high-level prediction errors reflect a greater need to revise internal models, suggesting that elevated Glx may support both associative learning and broader belief updating under uncertainty.”

Regarding the second point, we confirm that the high-level PEs used in the MRS analysis were unsigned, consistent with your understanding. In our task, it is the degree of unexpectedness of a stimulus, regardless of whether the outcome is “better” or “worse”, that drives learning. Unlike reinforcement / probabilistic reward learning tasks where outcomes have clear positive or negative valence, the probabilistic SRT task involves RT and accuracy, where the direction (sign) of the prediction error is less meaningful. As you

note, using unsigned PEs ensures that all deviations from expectation contribute to the measure, preventing positive and negative prediction errors from cancelling out.

[R4.13]

Regarding TA and uncertainty - while some past work has linked TA to uncertainty processing (Yan et al., 2025; Browning et al. 2015), a series of recent papers employing large sample sizes have not replicated these results (Satti et al., 2025; Suddell et al., 2024). The authors show a lack of relationship between TA and volatility, expected and unexpected uncertainty. I think adding a correlation matrix of these measures to supplementary would be helpful for the ongoing debate on the topic.

https://pure.mpg.de/rest/items/item_3592172/component/file_3634933/content
<https://doi.org/10.31234/osf.io/hm46n>

Thank you for this useful suggestion. We have now included a correlation matrix of these measures (page 9 of Supplement) and refer to it in the Discussion (Page 16, lines 298-304):

“Meanwhile, recent large-scale studies have questioned reports of altered uncertainty processing in anxiety^{29,30}. To contribute to this ongoing discussion, we conducted exploratory analyses examining relationships between trait anxiety and model-derived uncertainty estimates (volatility, expected, and unexpected uncertainty), none of which yielded significant correlations (Supplement). This suggests that the relationship between trait anxiety and belief-based uncertainty may be weaker or more context-dependent than previously assumed..”

Supplementary Figure S6. Correlation matrix showing associations between trait anxiety (STAI-Trait scores) and mean model-derived uncertainty measures: beliefs about volatility (ω), beta estimates for expected uncertainty ($\beta.U_{expected}$), and unexpected uncertainty ($\beta.U_{unexpected}$).

[R4.14]

Please state in Results which questionnaire was used to assess TA - I know it's in the Methods but it would help with readability.

We have now mentioned the specific questionnaire in the results sub-section pertaining to Trait Anxiety (Page 13, lines 229-231):

“We implemented an LME model on log-transformed RT with task session and Spielberger State-Trait Inventory (STAI) ⁵¹ trait anxiety scores as predictors, and STAI state anxiety scores as a controlling variable (LME Model 5).”

[R4.15]

Along similar lines, when reading the Results, I was missing basic information about i) the sample used and its size; ii) the general design of the study (this might give the impression that MRS was acquired simultaneously with the task).

We have now prefaced each of the results sub-sections (Page 7, Lines 120-122; Page 12, lines 196-197) with the required information for interpretation:

“We implemented a probabilistic serial reaction time (SRT) task in which participants (n=42) were told to indicate the location of a stimulus by pressing keys corresponding to four possible locations”

“After completing the behavioural task, participants completed a separate MRI session during which spectroscopy data were collected (Methods).”

We have also mentioned the study sample size in the figure captions.

[R4.16]

Exclusions - individual trials with RTs longer than 2 SDs were excluded (approx 4.5% of trials). What was the justification for this choice?

Thank you for highlighting this point. We excluded trials with RTs greater than 2 standard deviations above the mean to limit the influence of extreme outliers likely reflecting attentional lapses, momentary disengagement, or other non-task-related delays. Given that this task assessed implicit learning and required participants to simply view a stimulus and indicate its location, it involved minimal explicit cognitive demands. As such, extremely slow responses are unlikely to reflect meaningful variance in learning-related behaviour. This threshold is commonly employed in RT-based studies (e.g., Ratcliff, 1993; Whelan, 2008) to strike a balance between retaining meaningful variability and minimizing distortion in statistical or model-based estimates. Moreover, given the large number of trials (n = 1920), the proportion of excluded trials was relatively small and unlikely to have meaningfully impacted the results.

We have now added a brief justification of these exclusions to the Methods section (Page 21, lines 457-459):

“These exclusions were applied to minimise the influence of extreme values likely reflecting attentional lapses, momentary disengagement, or other non-task-related factors.”

- Ratcliff, R. (1993). Methods for dealing with reaction time outliers. *Psychological Bulletin*, 114(3), 510–532. <https://doi.org/10.1037/0033-2909.114.3.510>
- Whelan, R. (2008). Effective analysis of reaction time data. *The Psychological Record*, 58(3), 475–482. <https://doi.org/10.1007/BF03395630>

[R4.17]

Some references are in APA format, i.e. not numbered: Mathys 2011,2014; Weber 2024; Marshall 2016.

Thank you for pointing out these errors. We have corrected these.

[R4.18]

At least one point (p 14, line 227), the authors refer to “changes in volatility”. This is not accurate as there was only a single reversal.

We agree with this point and have omitted this phrasing in the revised manuscript. The sentence in question now reads as:

“We found that participants implicitly learned the probabilistic structure of a four-choice sensorimotor task and adapted to the probabilistic reversal.”

Computational signatures of uncertainty are reflected in motor cortex excitatory neurochemistry

Corresponding author: Nazia Jassim

Response to reviewers

We once again thank all four reviewers for the detailed and constructive feedback, which has contributed to improving the manuscript. Below, we provide a point-by-point response to each comment. Original reviewer comments are shown in blue and our responses in **bold black** font. Where required, revised text is indicated in *italics*, with page and line numbers provided for reference.

REVIEWERS' COMMENTS

Reviewer #1 (Remarks to the Author):

The authors have done an excellent job addressing the queries, the text now reads better and the new analyses (e.g. LME) strengthen the results, having a more principled way of assessing $\log(\text{RT})$ and surprise across trials than by using the (potentially arbitrary) three-way split of time into segments. I recommend the manuscript for publication.

We thank the reviewer for the positive feedback and are pleased that the revisions, which include the suggested LME analyses, have improved the clarity and strength of the manuscript.

Reviewer #1 (Remarks on code availability):

README file has improved.

We thank the reviewer for reviewing the README file and are pleased that they find it improved.

Reviewer #2 (Remarks to the Author):

The revisions made are extensive and have improved what was already an excellent paper. The authors should just double-check they've not introduced typos (e.g. should read 'evolution OF surprise' in response to R1.11), and 'three OF the comparison models' in response to R3.03.

We thank the reviewer for their positive feedback. We have carefully proofread the manuscript to ensure that no typos were introduced during the revision process, and have corrected the specific examples noted.

The rationale for the use of high-level delta was appreciated, thank you. It may be useful to relate surprise to Glx, too, as an exploratory analysis, and to aid our understanding of the specificity of the PE findings.

We thank the reviewer for this suggestion. To explore it, we conducted an additional correlation analysis between surprise and M1 Glx levels. This analysis revealed no significant association ($r = -0.21$, 95% CI $[-0.51, 0.14]$, $p = 0.24$), indicating that, unlike prediction errors, M1 Glx did not significantly relate to surprise in our dataset. Based on these results, we decided not to include this analysis in the manuscript in order to

maintain focus on the prediction error findings, which were both hypothesis-driven and theoretically grounded.

My question was also to what extent this study was preregistered - am I correct to assume it was not?

We thank the reviewer for raising this important point. The study was not formally preregistered. However, the central hypotheses that Glx levels would be associated with individual differences in adaptive learning signals derived from the HGF were specified *a priori* and motivated by prior work linking glutamatergic function to probabilistic learning. These confirmatory elements were thus hypothesis-driven. At the same time, we acknowledge that some analyses, including specific correlational analyses, were more exploratory in nature and have been described as such in the manuscript.

I am also satisfied with the explanation of the use of Glx not Glu/Gln separately. I think the inclusion of this explanation in the Discussion is also useful, given that I was not the only reviewer with this query! It may be worth briefly mentioning what indices you use of 'quantification uncertainty' in the text, too.

We thank the reviewer for their positive feedback and are pleased that the explanation regarding the use of Glx rather than Glu and Gln separately was clear. As suggested, we have also added a brief statement specifying the indices of quantification uncertainty used (Page 17, lines 349-373):

“Although 7T in principle offers improved spectral resolution, we chose to focus on Glx, as our data showed notable quantification uncertainty for glutamate and glutamine when estimated separately, and the two metabolites were weakly correlated. Specifically, this uncertainty was indexed by elevated Cramér-Rao lower bounds (CRLB), indicating less reliable estimates when attempting to separate the metabolites.”

Why are the colours so much darker in the novel correlation analysis without subjID? This implies the magnitudes of correlations have increased?

Thank you for this observation. The darker colours in the revised correlogram do not indicate an increase in correlation magnitudes. Rather, the colour scale in the figure was automatically rescaled after removing Subject ID from the input matrix, which changes the mapping between correlation values and colour intensity. The underlying correlation values between metabolites, including the key Glx–GABA correlation, remain essentially unchanged.

Thank you for clarifying the script locations. It would also be useful to link the two githubs together in some way – even by linking from one to the other in the 'readme' files.

We thank the reviewer for this helpful suggestion. We have now linked the two GitHub repositories in the README files to facilitate easier navigation between them.

Reviewer #3 (Remarks to the Author):

The authors have addressed my comments thoroughly. This is a very interesting paper that I am sure will be of interest to many, and I look forward to seeing it published.

We thank the reviewer for their positive feedback and are pleased that they found our revisions satisfactory. We are encouraged by their enthusiasm for the manuscript.

Reviewer #4 (Remarks to the Author):

Thank you for addressing the comments. I have no further suggestions or concerns.

We thank the reviewer for their feedback and are pleased that they have no further suggestions or concerns.

Reviewer #4 (Remarks on code availability):

I reviewed the code but I didn't run it. It looks sensible.

We thank the reviewer for taking the time to review the code.